# Allosteric regulation by c-di-AMP modulates a complete N-acetylglucosamine signaling cascade in *Saccharopolyspora erythraea*

Di You [1] ✉, Liu-Chang Zhao[1], Yu Fu[1], Zhi-Yao Peng[1], Zong-Qin Chen[1] & Bang-Ce Ye [1,2] ✉

c-di-AMP is an essential and widespread nucleotide second messenger in bacterial signaling. For most c-di-AMP synthesizing organisms, c-di-AMP homeostasis and the molecular mechanisms pertaining to its signal transduction are of great concern. Here we show that c-di-AMP binds the N-acetylglucosamine (GlcNAc)-sensing regulator DasR, indicating a direct link between c-di-AMP and GlcNAc signaling. Beyond its foundational role in cell-surface structure, GlcNAc is attractive as a major nutrient and messenger molecule regulating multiple cellular processes from bacteria to humans. We show that increased c-di-AMP levels allosterically activate DasR as a master repressor of GlcNAc utilization, causing the shutdown of the DasR-mediated GlcNAc signaling cascade and leading to a consistent enhancement in the developmental transition and antibiotic production in *Saccharopolyspora erythraea*. The expression of *disA*, encoding diadenylate cyclase, is directly repressed by the regulator DasR in response to GlcNAc signaling, thus forming a self-sustaining transcriptional feedback loop for c-di-AMP synthesis. These findings shed light on the allosteric regulation by c-di-AMP, which appears to play a prominent role in global signal integration and c-di-AMP homeostasis in bacteria and is likely widespread in *streptomycetes* that produce c-di-AMP.

Nucleotide signaling molecules are conserved, ubiquitous throughout all forms of life, and important secondary messengers in critical pathways that respond to environmental changes[1]. c-di-AMP is a notable secondary messenger identified in many gram-positive bacteria[2–4], many of which are prominent human pathogens and of environmental importance[5,6]. One of the most prominent functions of c-di-AMP is its regulatory role in cellular osmolarity and in controlling the transport of potassium and other osmolytes; accordingly, c-di-AMP is not essential in defined minimal media or rich media with lower salt concentrations[5,7,8]. A high extracellular concentration of potassium increases intracellular c-di-AMP levels, which limits potassium uptake and promotes potassium export[5]. In the absence of c-di-AMP, cells are not able to appropriately balance their osmotic state. Additionally, c-di-AMP participates in diverse essential processes, including cell wall and membrane homeostasis, DNA damage repair, biofilm formation, and immune response induction[6].

Many c-di-AMP-synthesizing organisms are prominent human pathogens and of environmental importance, which require c-di-AMP for growth and survival. A balanced level of c-di-AMP appears to be essential for normal bacterial physiology and virulence[9]. Given the importance it plays in diverse bacterial physiology, the molecular mechanisms pertaining to nutrient metabolism and the related signal transduction are of increasing focus. Components of the c-di-AMP signaling system have fundamental requirements to generate,

[1]Lab of Biosystems and Microanalysis, State Key Laboratory of Bioreactor Engineering, East China University of Science and Technology, Shanghai 200237, China. [2]Institute of Engineering Biology and Health, Collaborative Innovation Center of Yangtze River Delta Region Green Pharmaceuticals, College of Pharmaceutical Sciences, Zhejiang University of Technology, Hangzhou 310014 Zhejiang, China. ✉e-mail: 030111115@mail.ecust.edu.cn; bcye@ecust.edu.cn

degrade, and relay signals, thereby controlling bacterial growth and physiological functions[10]. Upon environmental stimulation, changes in intracellular c-di-AMP levels rely on the activity of diadenylate cyclase (DAC) domain-containing proteins or c-di-AMP-specific phosphodiesterase (PDE)[2]. In c-di-AMP-producing bacteria, signals that adjust the intracellular c-di-AMP concentration and the mechanism of this sensing, as well as the regulatory mechanism behind its synthesis and degradation, are poorly defined.

Although DACs and PDEs conform to a few well-conserved structural classes, the downstream targets of c-di-AMP signaling are structurally diverse, with multiple protein and riboswitches shown to bind c-di-AMP at different concentrations (KD values are in nM for the riboswitches and in µM for protein receptors)[2,10]. The identified c-di-AMP-binding effectors included Ktr/Trk potassium transporter family proteins containing RCK_C domains[11–14], osmolyte transporters such as OpuCA containing CBS domains[15], KdpD from several bacteria containing USP-like domains[16] and other domains[6]. There is no significant conservation among the c-di-AMP-binding motifs found in different c-di-AMP effectors, rendering bioinformatic identification impossible. Therefore, prediction or identification of novel c-di-AMP effectors, most notably transcription regulators, is very difficult.

In the current study, we show that the global regulator DasR, also the GlcNAc-sensing regulator, is a c-di-AMP effector. The binding of c-di-AMP by DasR directly links c-di-AMP to GlcNAc signaling. Accumulated evidence suggests that the amino sugar GlcNAc not only acts as a very good alternative carbon and nitrogen source[17–20] but also as a potent inducer of signaling to stimulate the expression of its own catabolic genes and morphogenetic switch, which plays a multifaceted role in cellular processes[20–22]. In addition, GlcNAc stimulates cellular signaling in pathogenic fungi such as *Candida albicans*[22]. For antibiotic-producing soil Actinobacteria, GlcNAc is a preferred carbon source, and its metabolite glutamate is even preferred over glucose[23,24].

Actinobacteria are aerobic gram-positive bacteria, and the most abundant organisms forming thread-like filaments in the soil. They are responsible for the characteristically "earthy" smell of freshly turned healthy soil and play major roles in the cycling of organic matter. Actinobacteria are of immense importance and the main source of antibiotics used in the human healthcare system[25]. Studies on GlcNAc signaling in Actinobacteria have provided new insight into the potential role of GlcNAc in evoking dynamic cellular responses. GlcNAc is a signal for antibiotic production and causes developmental arrest in antibiotic-producing soil Actinobacteria. The complete signaling cascade from the status of environmental GlcNAc to GlcNAc utilization[20,21,26–28] and the GlcNAc-triggered response was transmitted through the global pleiotropic regulator DasR[29,30]. During signaling, exogenous GlcNAc appears to adopt a simple mechanism of gene regulation by directly inactivating DasR, a GlcNAc sensor and transducer. Glucosamine-6-phosphate, the metabolic intermediate of GlcNAc, allosterically impairs the DNA-binding ability of DasR towards its target promoters to activate the transcriptional response involved in GlcNAc catabolism[27]. The GntR-family regulator DasR functions as the developmental master regulator, and is predominantly a dimer and a pleiotropic regulator that oversees the interplay between primary and secondary metabolism[20,27,28].

In this work, we identify DasR of erythromycin-producing *Saccharopolyspora erythraea* as a target of c-di-AMP, thus revealing a direct link between c-di-AMP and GlcNAc signaling in this bacterium. We demonstrate that c-di-AMP binding stimulates DasR-mediated repression of GlcNAc uptake and utilization, and describe the molecular mechanism by which c-di-AMP mediates the formation of a c-di-AMP-linked DasR dimer. Since GntR-family regulators interact with DNA as dimers[31], such effective dimerization of DasR activates DasR-mediated regulation of its target regulon. Specifically, DAC activity is under direct transcriptional regulation by DasR, thus forming a

regulatory circuit that governs the complete signaling cascade from nutritional status to the onset of morphological differentiation and antibiotic production. Finally, our phylogenetic analysis and in vitro assays further indicate that allosteric regulation by c-di-AMP exerts global regulation, and signal integration mediated through DasR is likely conserved and essential among c-di-AMP-producing Actinobacteria.

## Results

### c-di-AMP impairs GlcNAc utilization and the GlcNAc-triggered response

c-di-AMP is synthesized by the DAC activity of DisA_N domain-containing proteins (Pfam PF02457)[10], which are present in more than 11,000 organisms. To gain insight into the cellular processes controlled by c-di-AMP in gram-positive *S. erythraea*, we overexpressed its only DAC DisA and obtained the O*disA* strain. With the WT strain used as a control, the transcription levels of *disA* and intracellular c-di-AMP in the two strains grown in a liquid TSB medium were measured. As shown in Fig. 1a, *disA* overexpression caused a nearly 20-fold increase in *disA* transcription compared with that in the WT strain, and the intracellular c-di-AMP in cell extracts was consistent with *disA* transcription, showing a more than five-fold increase in the intracellular c-di-AMP level of the O*disA* strain in comparison to that of the WT strain. We first applied different culture media to profile bacterial behavior affected by c-di-AMP levels. GlcNAc or glucose addition was used for investigation. There were no obvious growth differences observed when the strains were cultured in glucose (Fig. 1b), whereas the O*disA* strains presented significant growth inhibition compared to the WT strain in presence of GlcNAc (Fig. 1c). The GlcNAc consumption in the two strains was then determined and indicated that DisA overexpression substantially inhibited the utilization of GlcNAc (Fig. 1c). GlcNAc has been established to block development and antibiotic production under rich growth conditions[20]. To assess whether c-di-AMP affects this GlcNAc-induced inhibitory effect, WT and O*disA* strains were plated onto R2YE agar with GlcNAc or glucose addition. Notably, DisA overexpression caused accelerated differentiation and sporulation compared with the WT strain in glucose (Fig. 1d), and the O*disA* strain also showed unperturbed sporulation even in presence of GlcNAc (Fig. 1e). Detailed scanning electron microscopy (SEM) examination further revealed that the O*disA* strain contained developed mature spores bypassing GlcNAc-triggered inhibition (Fig. 1f, g). Accordingly, the influence of DisA overexpression on antibiotic production showed that overexpression of DisA led to consistently enhanced erythromycin production under both conditions (Fig. 1h, Supplementary Fig. 1), indicating that c-di-AMP had a stimulating effect on antibiotic biosynthesis in *S. erythraea*. These data revealed that the intracellular level of c-di-AMP influences development and antibiotic biosynthesis. In particular, the results suggested that increased c-di-AMP levels accelerated sporulation and arrested bacterial GlcNAc utilization and its cascading effect on secondary metabolism, laying the groundwork for detailed characterization of the impact of intracellular c-di-AMP.

### The GlcNAc-sensing protein DasR is a c-di-AMP effector

GlcNAc sensing and the response transmitted to development and antibiotic production are controlled by the pleiotropic transcriptional regulator DasR[21,26], which raised the possibility that DasR might be a potential c-di-AMP receptor protein. To probe this interaction, His-DasR with c-di-AMP, c-di-GMP, cAMP, or cGMP was subjected to a biolayer interferometry (BLI) assay. As shown in Fig. 2a, b, an increase in the amount of c-di-AMP was accompanied by a corresponding increase in the response. In contrast, no significant change in the response was observed with c-di-GMP, cAMP, or cGMP. Examination of isothermal titration calorimetry (ITC) experiments with c-di-AMP and DasR showed that the dissociation constant (KD) was 26.8 µM

(Fig. 2c, Supplementary Fig. 2). Further confirming our inference, DasR specifically binds c-di-AMP, and DasR from *S. erythraea* is a c-di-AMP effector.

## c-di-AMP remarkably enhances the DNA-binding activity of DasR in vitro

DasR functions as a transcriptional regulator through binding with its target promoters. To investigate the effect of c-di-AMP on the DNA-binding activity of the DasR regulator, the promoter of the DasR target gene *nagA*[21,32,33] was selected and subjected to the electrophoretic mobility shift assay (EMSA) with DasR in the absence or in the presence of c-di-AMP. The results showed that in the absence of c-di-AMP, DasR formed a relatively stable complex with DNA fragments at a protein concentration of 20 μM, as evidenced by a clear mobility shift (Fig. 2d, top); the DNA-binding activity of DasR was enhanced in the presence of c-di-AMP, such that DasR formed stable complexes with DNA fragments at a low protein concentration (2 μM) (Fig. 2d, bottom). Additionally, BLI also confirmed that DasR and DNA had a binding affinity of ~19 μM in the absence of c-di-AMP, whereas the KD was ~0.44 μM, a more than 40-fold increase, after preincubation with c-di-AMP (Fig. 2e). Further analysis supported that this enhancement also works with lower concentrations of c-di-AMP (Supplementary Fig. 3).

These results indicated that c-di-AMP strongly induced the DNA-binding ability of DasR.

GntR-family regulators interact with DNA as dimers[31]. To determine whether c-di-AMP binding influences DasR dimerization, circular dichroism (CD) and chemical cross-linking assays were performed. The far-UV CD spectra showed an increase in ellipticity at 195-220 nm, indicating a decrease in the α-helical content of the DasR-c-di-AMP complex and suggesting that c-di-AMP altered the secondary structure of DasR (Fig. 2f). The DasR-c-di-AMP complex showed significantly lower α-helicity and a concomitant increase in antiparallel and random coil structures compared with the secondary structure of DasR alone (Fig. 2f, Supplementary Table 1). The oligomeric state of DasR examined by chemical cross-linking clearly showed that multiple bands with low mobility were observed in the presence of c-di-AMP, and c-di-AMP induced the formation of DasR dimers in a concentration-dependent manner (Fig. 2g). Hence, the cross-linking and CD data supported the notion that c-di-AMP induces dimerization of DasR. To characterize the putative c-di-AMP-DasR complex, mutagenesis experiments and molecular docking were conducted. The molecular docking results suggested that the binding of c-di-AMP occurred in the effector-binding (EB) domain of DasR (Supplementary Fig. 4a, b). Generally, the helix-turn-helix (HTH) motif mediates DNA binding. To probe the

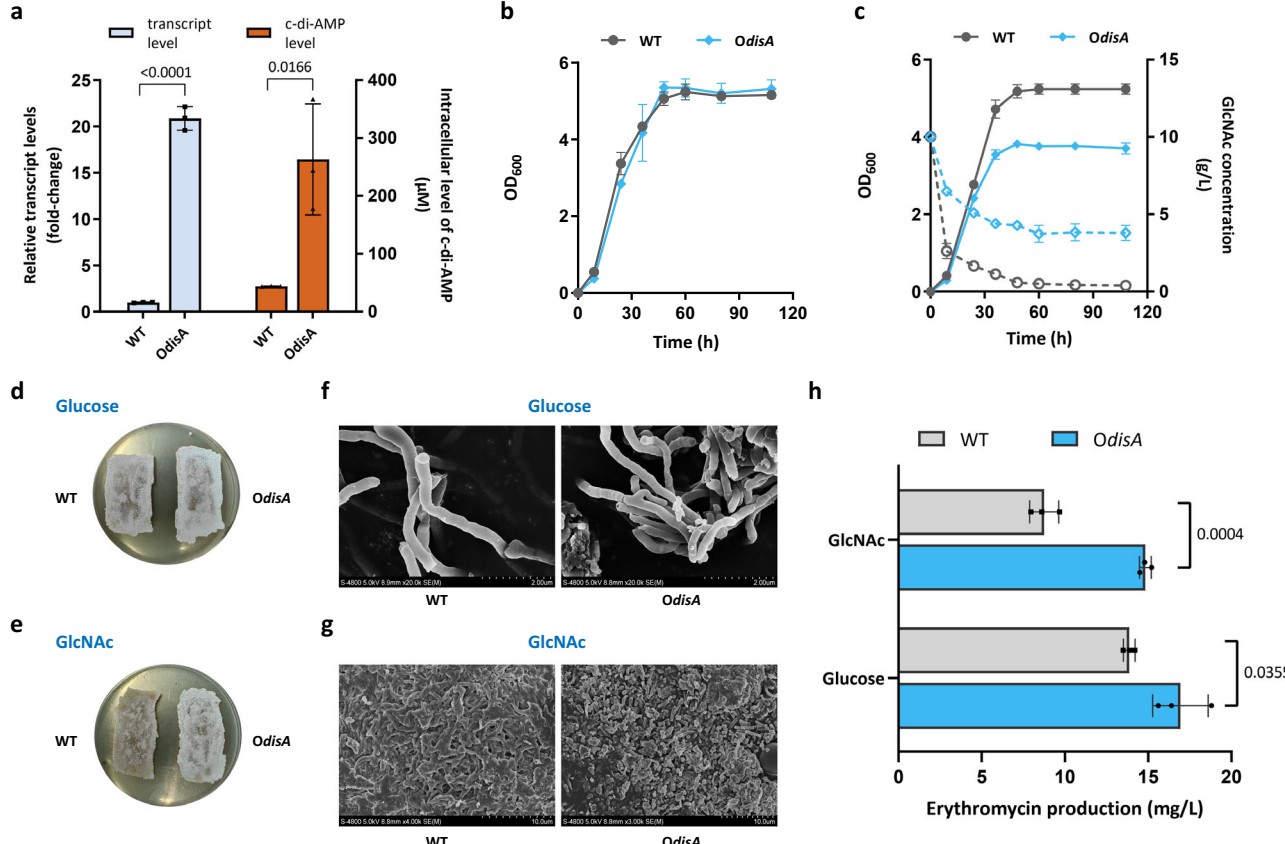

**Fig. 1 | c-di-AMP impairs GlcNAc utilization and the GlcNAc-triggered response.** **a** *disA* transcription levels and intracellular c-di-AMP concentrations in WT and O*disA* strains grown in liquid TSB medium during late exponential growth (48 h). Fold change represents the expression level or c-di-AMP level compared to the WT strain. The c-di-AMP concentrations of the samples were normalized to the dry cell weight. Data are presented as mean values ± SD for *n* = 3 biological replicates. **b** Growth curves of *S. erythraea* WT and O*disA* strains grown at 30 °C in liquid TSB medium with glucose addition. Data are presented as mean values ± SD for *n* = 3 biological replicates. **c** Growth curves and GlcNAc utilization of *S. erythraea* WT and O*disA* strains grown at 30 °C in liquid TSB medium with GlcNAc addition. Solid lines indicate growth and dashed lines illustrate GlcNAc utilization. Data are presented as

mean values ± SD for *n* = 3 biological replicates. Phenotypes of WT and O*disA* strains grown for 168 h on nutrient-rich R2YE agar plates with glucose (**d**) or GlcNAc addition (**e**). Representative pictures of two independent experiments with similar results are shown. SEM examination of WT and O*disA* strains grown for 72 h on R2YE agar plates with glucose (**f**) or GlcNAc addition (**g**). Representative pictures of two independent experiments with similar results are shown. **h** Quantitative analysis of erythromycin production of WT and O*disA* strains by HPLC. Erythromycin was extracted from cultures grown for 120 h in 50 ml TSB at 30 °C. Data are presented as mean values ± SD for *n* = 3 biological replicates. An unpaired two-sided *t* test was used for the statistical analysis. Source data are provided as a Source Data file.

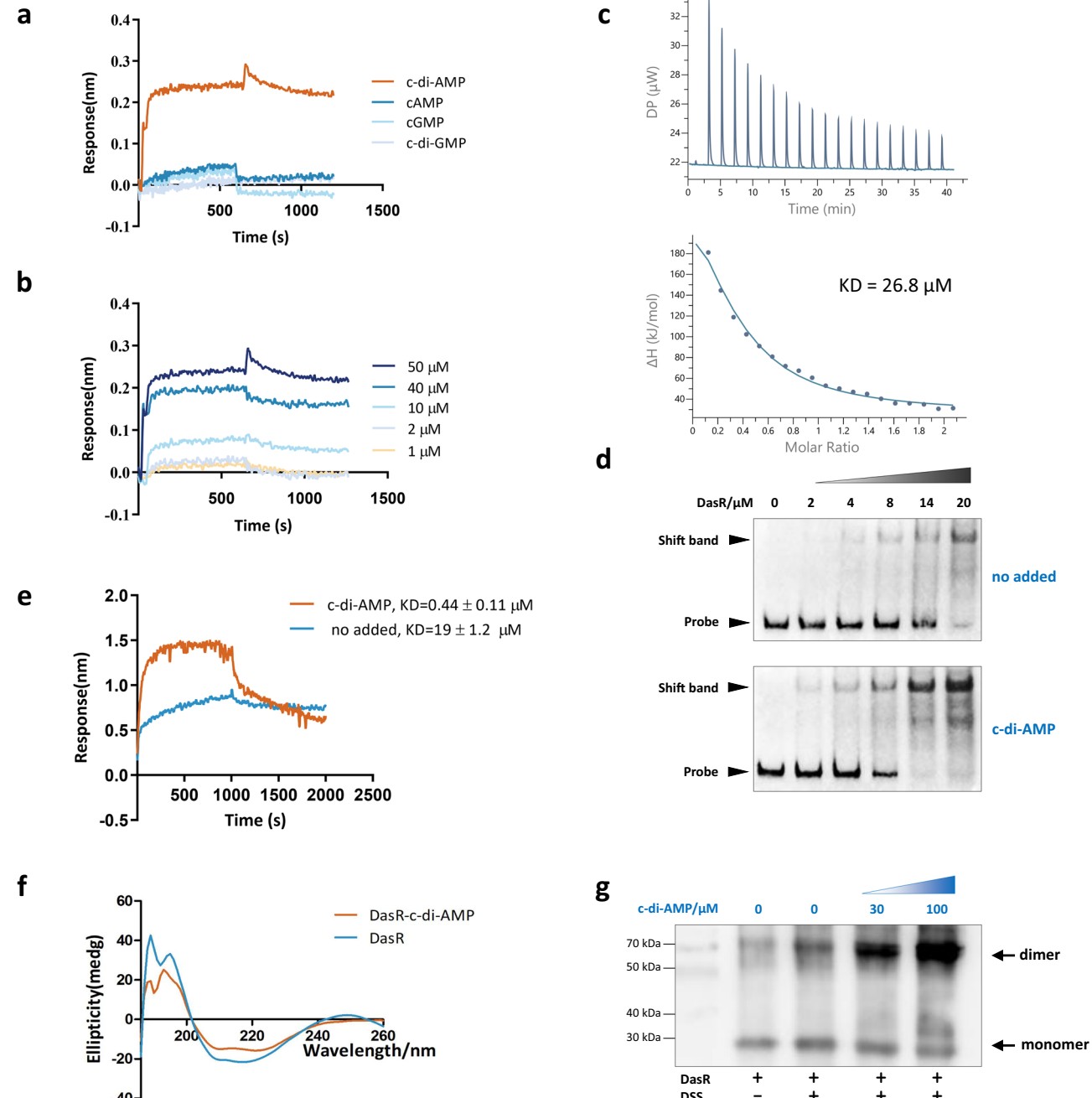

**Fig. 2 | c-di-AMP is an allosteric activator of DasR−DNA binding. a** Biolayer interferometry assay of purified His-DarR with 100 μM c-di-AMP, c-di-GMP, cAMP, or cGMP. **b** Biolayer interferometry assay of purified His-DarR with a gradient of c-di-AMP. **c** Characterization of the interaction of c-di-AMP with DasR using ITC. Titration of c-di-AMP (300 μM) into DasR (30 μM). **d** EMSA of DasR binding to its target gene *nagA* promoter. Purified His-DasR and DNA fragments were incubated with 100 μM c-di-AMP (bottom) or without c-di-AMP (top). Representative pictures of two independent experiments with similar results are shown. **e** Biolayer interferometry assay of purified His-DarR and DNA (*nagA* promoter) in the presence of 100 μM c-di-AMP or in the absence of c-di-AMP. **f** Circular dichroism spectra of DasR before or after incubation with c-di-AMP. **g** Cross-linking with a gradient concentration of c-di-AMP (0, 30, and 100 μM). Samples were analyzed by SDS-PAGE. Monomers and dimers are marked by arrows. Representative pictures of two independent experiments with similar results are shown. Source data are provided as a Source Data file.

binding mechanism further, we created mutations (Supplementary Fig. 4c) that retained binding at the HTH motif but disabled binding at the EB motif. Based on our data (Supplementary Fig. 4d), DasR-HTH bound to DNA, whereas no reliable DNA-binding activity was detected for DasR-EB. We also observed that an EB deletion mutant of DasR was incapable of super binding in the presence of c-di-AMP (Supplementary Fig. 4d). Thus, the EB motif of DasR appears to be the main motif

responsible for its ability to bind c-di-AMP. The putative DasR-c-di-AMP docking analysis (Supplementary Fig. 4a, b) also suggested that c-di-AMP acted as a dimerizer to link two DasR protomers to drive DasR dimerization. Together, these results demonstrated that c-di-AMP functions as an allosteric activator for DasR−DNA binding, which could facilitate the formation of a more stable complex and exert transcriptional regulation on the DasR regulon in vivo.

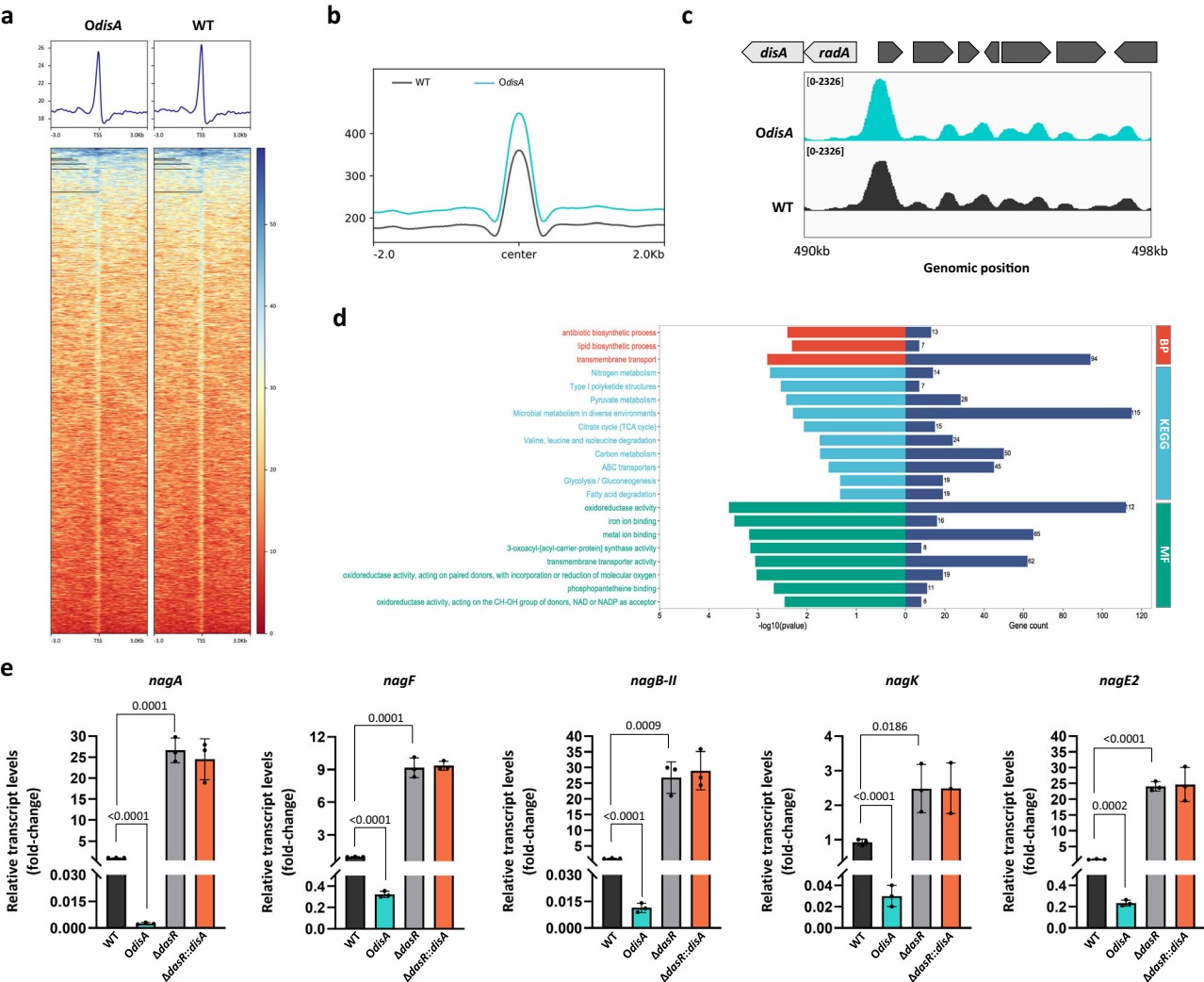

**Fig. 3 | c-di-AMP stimulates DasR-mediated repression of its target regulon in vivo. a** Heatmaps of the ChIP-seq signal density at the peak center and TSSs (±3 kb). The average signal profile is shown. The red color indicates a low signal, and the blue color indicates a high signal. **b** ChIP-seq signal in the indicated strains. **c** Integrative Genomics Viewer (IGV) tracks showing ChIP-seq signals at the promoter regions of *disA* in the indicated strains. **d** GO terms and KEGG analysis enriched for upregulated target genes in the network. **e** The transcription levels of the indicated genes in the *S. erythraea* WT, O*disA*, Δ*dasR,* and Δ*dasR::disA* strains grown in liquid TSB medium during late exponential growth (48 h). Fold change represents the transcription level compared to the WT strain. Data are presented as mean values ± SD for *n* = 3 biological replicates. Ordinary one-way ANOVA was used for the statistical test. Source data are provided as a Source Data file.

## c-di-AMP induces the global outcome of DasR binding to its targeted genes

To confirm and extend the allosteric regulation by c-di-AMP into cells, we used chromatin immunoprecipitation sequencing (ChIP-seq) to determine the effects of elevated c-di-AMP on DasR-dependent binding genome-wide in *S. erythraea*. Wild-type *S. erythraea* and the DisA overexpression strain were grown in TSB liquid and subjected to ChIP-seq experiments using a specific anti-DasR antibody. The total (non-immunoprecipitated) input DNA from each strain was also subjected to sequencing. Both WT and O*disA* signals were widely distributed at transcription start sites (TSSs) with a sharp single peak (see Fig. 3a). A total of ~99% of the detected peaks in the O*disA* strain showed enhanced ChIP-seq peak height at the DasR target promoters relative to the WT control, confirming the elevated occupancy in the O*disA* strain (Fig. 3b). Collectively, the representative results following visualization and verification of the related genes during exposure to high c-di-AMP levels illustrated changes in the ChIP-seq peaks at the individual gene level (Fig. 3c). GO and KEGG pathway analyses of the genes with enhanced binding signals indicated enrichment in the antibiotic biosynthetic process;

carbon, nitrogen and pyruvate metabolism; and microbial metabolism in diverse environments (Fig. 3d). These results further demonstrated that c-di-AMP indeed enhances the binding of DasR to its targeted genes in vivo. DasR is a GlcNAc sensor and master regulator in GlcNAc signaling that directly represses GlcNAc catabolic gene expression. To directly examine this enhancement towards DasR-mediated repression, GlcNAc assimilation-related genes, including *nagK*, *nagB-II*, *nagA*, *nagF*, and *nagE2*, were then verified by real-time RT-PCR monitoring of transcription. Consistent with the ChIP-seq data, the repression of the selected genes was strongly enhanced in O*disA* (Fig. 3e). However, the results in the Δ*dasR::disA* strain showed that c-di-AMP had no effect on the transcription of GlcNAc assimilation-related genes in the *dasR*-deficient condition, supporting that c-di-AMP through DasR controls the GlcNAc assimilation. The observed data validated the previous results showing that excess c-di-AMP serves to turn off GlcNAc assimilation, thus impairing GlcNAc signal transduction to morphological development and secondary metabolite production. This analysis helps annotate a relatively uncharacterized role of c-di-AMP by its functional association with DasR.

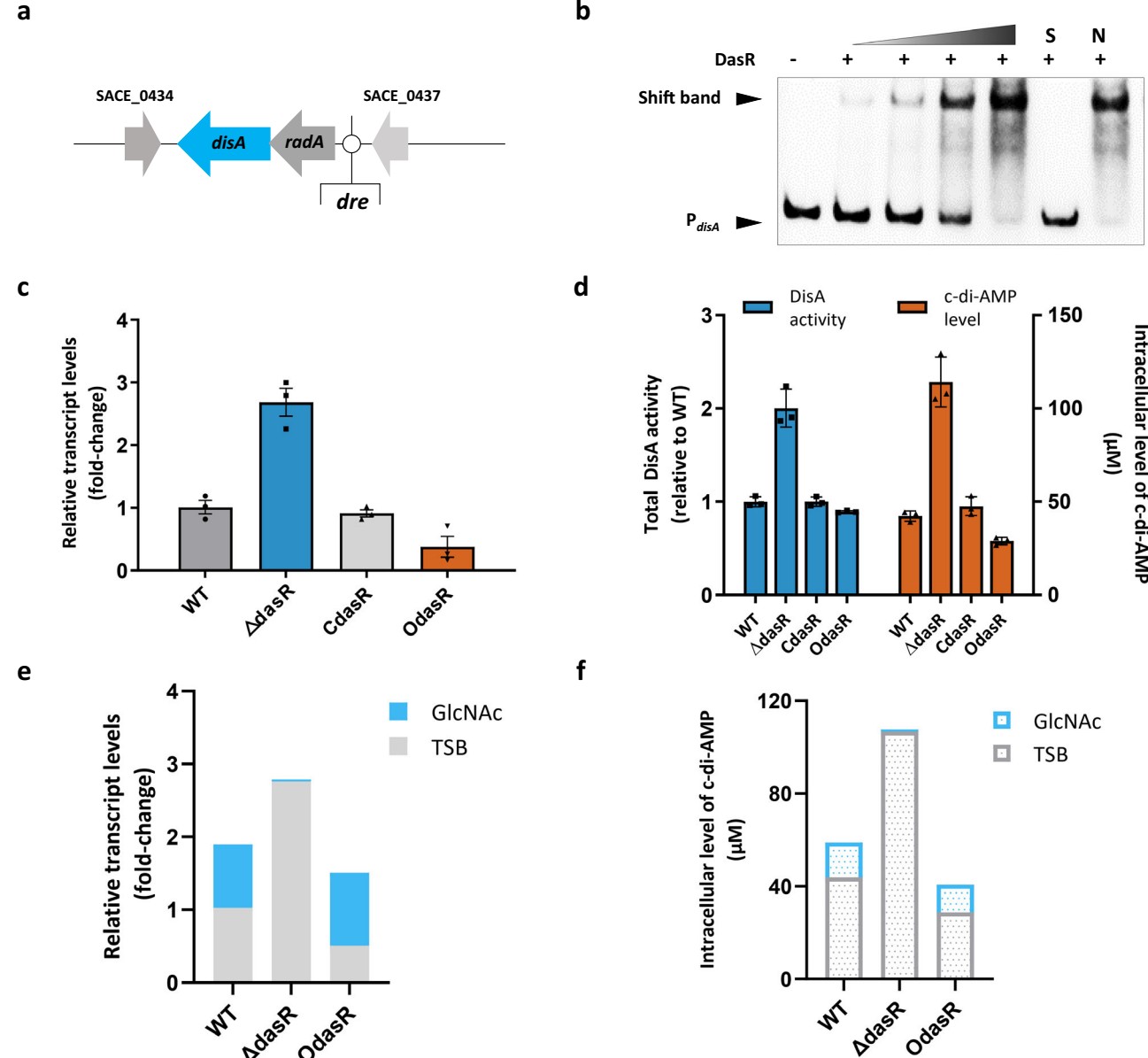

**Fig. 4 | DasR and GlcNAc influence c-di-AMP synthesis in *S. erythraea*. a** DasR-responsive elements (*dre*) in the *disA* promoter region. **b** EMSA of purified DasR with the *disA* promoter region. DNA probes (10 nM) were incubated with a gradient concentration of purified DasR. EMSAs with a 200-fold excess of unlabeled specific probe (S) or non-specific competitor DNA (sperm DNA) (N) were used as controls. Representative pictures of two independent experiments with similar results are shown. **c** The *disA* transcription levels in WT, Δ*dasR*, C*dasR*, and O*dasR* strains grown in liquid TSB medium during late exponential growth (48 h). Fold change represents the expression level compared to the *disA* expression in the WT strain. Data are presented as mean values ± SD for *n* = 3 biological replicates.
**d** Intracellular c-di-AMP concentrations and total DisA DAC activity in cell extracts of *S. erythraea* WT, Δ*dasR*, C*dasR*, and O*dasR* strains grown in liquid TSB medium during late exponential growth (48 h). The c-di-AMP concentrations of the samples were normalized to the dry cell weight. Data are presented as mean values ± SD for *n* = 3 biological replicates. **e** GlcNAc induces *disA* transcription through DasR. The *disA* transcription levels in WT, Δ*dasR*, and O*dasR* strains grown in liquid TSB medium with or without GlcNAc addition during late exponential growth (48 h). Fold change represents the expression level compared to the *disA* expression in the WT strain grown in liquid TSB medium without GlcNAc addition. Blue bars show the increase caused by GlcNAc addition. Data are presented as mean values ± SD for *n* = 3 biological replicates. **f** GlcNAc induces c-di-AMP accumulation through DasR. Intracellular c-di-AMP concentrations in *S. erythraea* WT, Δ*dasR*, and O*dasR* strains grown in TSB medium with or without GlcNAc addition during late exponential growth (48 h). The c-di-AMP concentrations of the samples were normalized to the dry cell weight. Blue bars show the increase caused by GlcNAc addition. Data are presented as mean values ± SD for *n* = 3 biological replicates. Source data are provided as a Source Data file.

## DasR influences intracellular c-di-AMP levels via direct transcriptional repression of *disA*

Intriguingly, *disA* was identified as a potential gene target of DasR according to ChIP-seq analysis (Fig. 3c). DasR-responsive elements (*dre*) in Actinobacteria[20,26,28] were identified in previous studies, and more than five putative *dre* sites were found in the region upstream of the *disA* gene in *S. erythraea* (Fig. 4a). To confirm whether DasR could

directly bind the region upstream of *disA*, EMSA was performed. Obvious band shifts were observed following incubation with purified His-tagged DasR (Fig. 4b), suggesting that DasR specifically bound the *disA* promoter region. We then examined whether DasR has a regulatory effect on *disA* using the wild-type (WT) *S. erythraea* strain, a *dasR*-deletion strain (Δ*dasR*), a *dasR*-complemented strain (C*dasR*), and a WT strain overexpressing *dasR* (O*dasR*) constructed as described

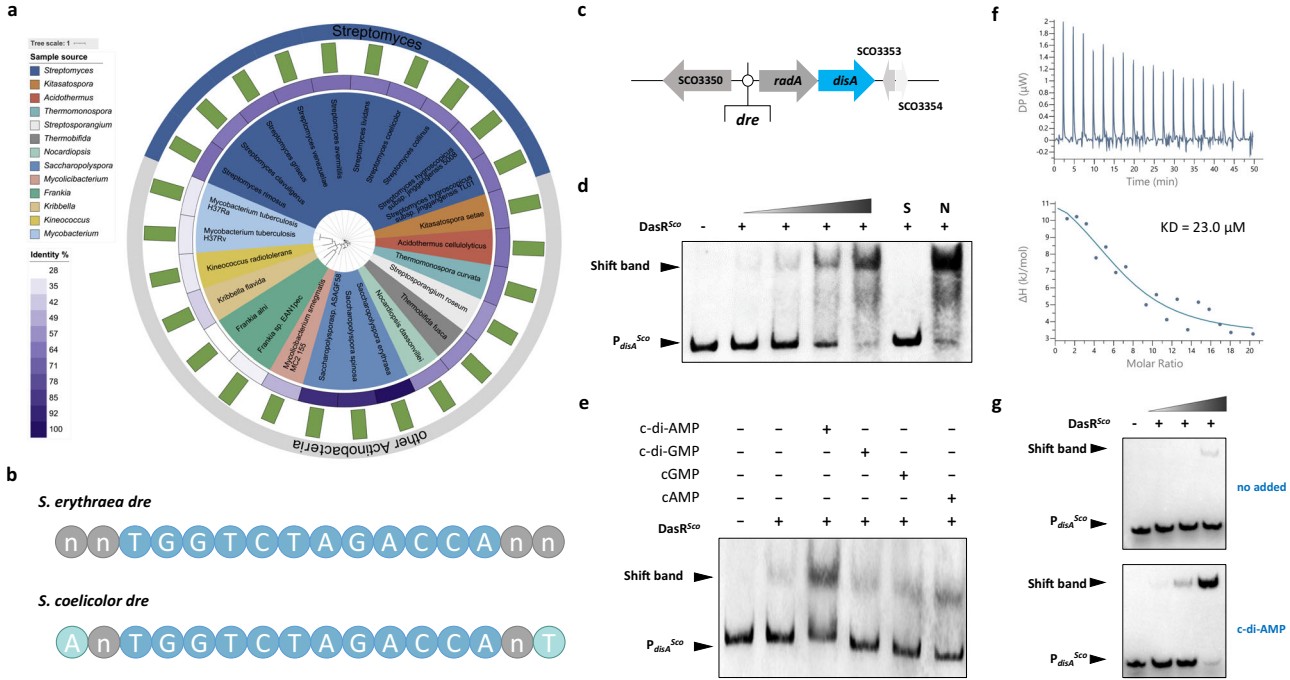

**Fig. 5 | The interaction of c-di-AMP and DasR is conserved in *S. coelicolor*.**
**a** Phylogenetic analysis of the DasR protein in prokaryotes. Sequences of DasR proteins were aligned with MAFFT v7.489 with L-INS-I. Trees were reconstructed from the resulting alignments using FastTree and subsequently visualized inter-active Tree of Life (iTol) Version 6.5.6. The green bar represents the aa length of the DasR protein. Left panels show the genus names analyzed in the phylogenetic survey and the identity of the *S. erythraea* DasR sequence aligned with the corresponding homologs. **b** DasR-responsive elements (*dre*) elements of *S. coelicolor* DasR and *S. erythraea* DasR. **c** *dre* in the *disA* promoter region of *S. coelicolor*. **d** EMSA of purified DasR with the *disA* promoter region of *S. coelicolor*. DNA probes (10 nM) were incubated with a gradient concentration of purified DasR. EMSAs with a 200-fold excess of unlabeled specific probe (S) or non-specific competitor DNA (sperm DNA) (N) were used as controls. Representative pictures of two independent experiments with similar results are shown. **e** EMSA of c-di-AMP specifically stimulates the DasR–DNA binding of *S. coelicolor*. Representative pictures of two independent experiments with similar results are shown. **f** Characterization of the interaction of c-di-AMP with DasR$^{Sco}$ using ITC. Titration of c-di-AMP (1 mM) into DasR$^{Sco}$ (10 μM). **g** EMSA of DasR$^{Sco}$ binding to its target gene *disA* promoter. Purified His-DasR$^{Sco}$ and DNA fragments were incubated with 30 μM c-di-AMP (bottom) or without c-di-AMP (top). Representative pictures of two independent experiments with similar results are shown. Source data are provided as a Source Data file.

previously[21,30]. The transcription level of *disA* in the four strains was detected by real-time RT–PCR. As shown in Fig. 4c, *dasR* deficiency caused a ~three-fold increase in *disA* transcription compared with that in the WT strain, *disA* transcription in the *dasR*-complemented strain was similar to that in the WT strain, and the WT strain overexpressing *dasR* showed a ~60% decrease in *disA* expression. Hence, DasR directly repressed *disA* transcription in *S. erythraea*.

Bacterial c-di-AMP synthesis is catalyzed by the DAC activity of DisA. To investigate whether DasR affects c-di-AMP synthesis in *S. erythraea*, we determined the intracellular c-di-AMP levels and total DisA DAC activity in the WT, Δ*dasR*, C*dasR*, and O*dasR* strains. A more than twofold increase in the intracellular c-di-AMP level of the Δ*dasR* strain was observed in comparison to that of the WT strain, and the O*dasR* strain showed a more than 40% decrease in c-di-AMP synthesis (Fig. 4d). Additional evidence that c-di-AMP synthesis was directly regulated by DasR was provided by the measurement of total DisA DAC activity in cell extracts using HPLC. The results were consistent with the intracellular c-di-AMP levels, showing that *dasR* deficiency caused a 2-fold increase in DisA DAC activity (Fig. 4d). Therefore, together with the observation that c-di-AMP induced allosteric activation for DasR–DNA binding, the combined data suggest an autofeedback loop working model of the intracellular c-di-AMP pool.

### GlcNAc stimulates c-di-AMP accumulation in a DasR-dependent manner

To explore the utility of the GlcNAc signal for c-di-AMP synthesis, we further evaluated the transcription of *disA* and intracellular c-di-AMP levels in *S. erythraea* WT, Δ*dasR*, and O*dasR* strains cultured in TSB medium with or without GlcNAc addition. The *disA* transcription level of the WT strain grown with GlcNAc revealed a two-fold increase compared to TSB medium, and the O*dasR* strain grown with GlcNAc had a more obvious increase in *disA* transcription (Fig. 4e). However, GlcNAc had no effect on *disA* transcription in the *dasR*-deficient strain (Fig. 4e). In parallel, the intracellular c-di-AMP concentration showed a consistent trend, supporting the idea that GlcNAc signals through DasR to control the intracellular c-di-AMP level (Fig. 4f). Taken together, these results indicated that c-di-AMP synthesis is induced by GlcNAc through the release of DasR-mediated repression. Thus, we have outlined a mode of c-di-AMP working in concert with DasR, which dominates bacterial GlcNAc, triggering important cellular processes.

### Allosteric regulation between c-di-AMP and DasR is conserved in *S. coelicolor*

DasR is crucial across Actinobacteria and exhibits high sequence similarity with its homologs[28]. To complement these evolutionary insights, the *S. erythraea* DasR sequence was aligned with homologs in other c-di-AMP-producing species, including SCO5231 from *S. coelicolor*, MSMEG_0286 from *Mycobacterium smegmatis*, and RV0792c from *Mycobacterium tuberculosis*, etc. (Fig. 5a). Phylogenetic survey revealed that DasR homologs are broadly distributed among c-di-AMP synthesizing species, particularly for Streptomyces species. We, therefore, speculated that DasR as a c-di-AMP receptor would be comparatively conserved in *streptomyces* and some other genera of Actinobacteria. DasR of *S. erythraea* was highly homologous to that of *S. coelicolor*, with an identity of 72%, and *dre* also exhibited high sequence similarity (Fig. 5b). Similar to observations for *S. erythraea*,

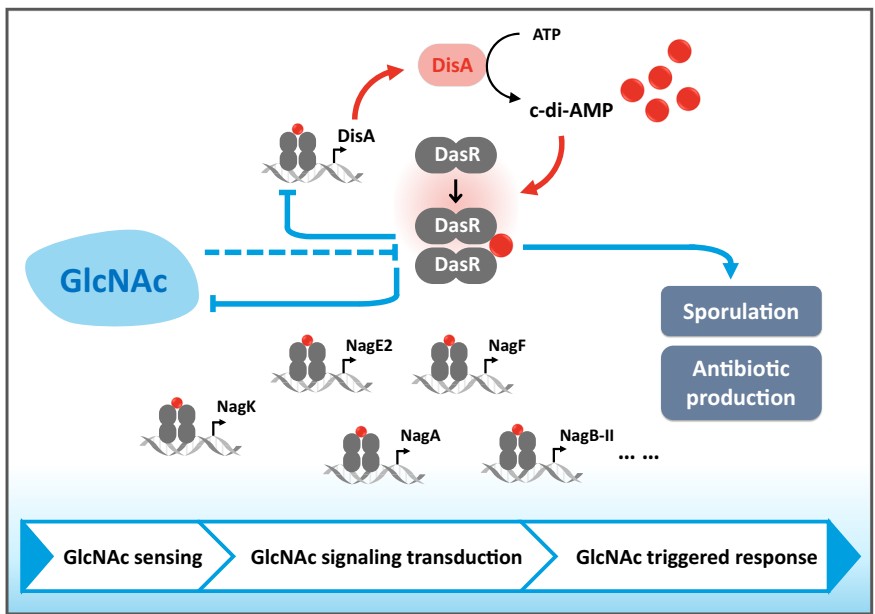

**Fig. 6 | Working model for the cross-talk between c-di-AMP and GlcNAc signaling.** This model summarizes the changes in c-di-AMP levels during changes in GlcNAc availability and their signaling effect on DasR activity, thus revealing the regulatory cascade downstream of DasR that controls the development and antibiotic biosynthesis. The arrows indicate positive controls. The perpendicular lines indicate the negative control.

multiple *dre* sites were found in the region upstream of the *radA/disA* cluster in *S. coelicolor* (Fig. 5c). To verify the possible interaction between *S. coelicolor* DasR and c-di-AMP, we investigated the effect of *Sco*DasR on *disA* transcriptional regulation and the allosteric activation function of c-di-AMP in DasR–DNA binding. *Sco*DasR was found to directly interact with the region upstream of the corresponding *disA* gene (Fig. 5d). In addition, *Sco*DasR–DNA binding increased in the presence of c-di-AMP, whereas in the presence of other nucleotides, such as c-di-GMP, cGMP and cAMP, there were little differences observed in DasR–DNA binding (Fig. 5e). We also performed ITC assays using *Sco*DasR titrated with c-di-AMP, cAMP, cGMP, and c-di-GMP. A specific interaction between c-di-AMP and the *Sco*DasR protein was also observed, with a KD of 23.0 µM (Fig. 5f), and none of the other detected nucleotides bind *Sco*DasR (Supplementary Fig. 5). These observations support a specific interaction with c-di-AMP. Moreover, the EMSA (Fig. 5g) led to the same conclusion as the one using DasR from *S. eryhraea*: c-di-AMP strongly induced *Sco*DasR binding to DNA. Taken together, these results show that DasR is a bona fide bacterial c-di-AMP receptor protein and support a model whereby c-di-AMP enhances the DNA-binding activity of DasR in *S. coelicolor*, further supporting that allosteric regulation by c-di-AMP might be a major mechanism common in c-di-AMP-producing Actinobacteria.

## Discussion

The role of c-di-AMP has been studied extensively in the maintenance of osmotic pressure, biofilm formation, acid stress resistance, the response to DNA damage, and other functions in gram-negative bacteria[6]. Variations in the c-di-AMP level (both high and low c-di-AMP levels) cause metabolic imbalance, which is a signal altering cell proliferation or related metabolic pathways. To obtain a more detailed view of the molecular mechanism underlying c-di-AMP signaling, a series of c-di-AMP-specific receptors, including riboswitches and proteins, that allow this cyclic dinucleotide to control bacterial physiology were identified. c-di-AMP receptors are typically flexible in conformation, which enables a variety of physiological functions such as osmotic regulation, metabolic response, and developmental regulation. Among the protein receptors of c-di-AMP, most are not of a single domain family, and the conservation of receptors appears to be relatively poor, even among related organisms of the same phylum. In the current study, we showed that the activity of the master regulator DasR, also the GlcNAc-sensing regulator, is controlled by c-di-AMP. DasR is a global regulator and an example of a GntR-family regulator identified as the c-di-AMP receptor protein, which is quite distinct from the two c-di-AMP-binding transcriptional regulators[34,35] characterized to date. DasR is a GntR member belonging to the HutC subfamily whose members have been implicated in processes like antibiotic production and as sensors of the nutritional status in the environment. Moreover, DasR homologs are widespread among hundreds of microorganisms synthesizing c-di-AMP. It is, therefore, likely that DasR represents a broadly conserved class of c-di-AMP receptors, bringing the regulatory role of this key second messenger into a new physiological arena in gram-positive multicellular Actinobacteria.

Our studies indicate that DasR negatively contributes to the intracellular c-di-AMP pool via direct repression of *disA* transcription. This illustrates that, in addition to the catalytic enzymes that control intracellular c-di-AMP homeostasis, transcriptional regulators are also vital in regulating intracellular c-di-AMP levels. Specifically, we demonstrate that c-di-AMP binding to DasR activates its DNA-binding activity, contributing to an enhancement of the DasR regulon. This negative feedback loop presents a self-sustaining mode as follows: Once the c-di-AMP levels are exceeded, the feedback loop is activated to repress its own synthesis, thus avoiding its toxic accumulation (Fig. 6). The cross-talk between c-di-AMP and DasR forms a homeostatic signal loop to maintain appropriate c-di-AMP states within the bacterial cell, exhibiting therapeutic or industrial potential for preventing microbial growth. Most bacteria encode only one or a few DACs[1], and c-di-AMP is essential in many but not all bacterial species. The essentiality of c-di-AMP was dependent on the growth conditions[6]. Thus, the interdependencies between c-di-AMP and DasR conceivably require a high degree of coordination and involve numerous different cellular processes. Of note, the implications of this model may differ between the species and the host specificity of the effect of c-di-AMP requires further exploration. In addition, the structural determination of the c-di-AMP-DasR complex will be interesting to investigate in the future.

Transcription factors bind to specific sequences in the genome to alter gene expression and specify cell states[36]. DasR is a key developmental regulator that coordinates developmental transition and antibiotic production, which oversees the entire GlcNAc signaling cascade through the repression of a global regulon of the related genes[20,21,26,27]. Apart from its structural functions, GlcNAc stimulates cellular signaling in bacterial development, and the production of antimicrobial compounds makes it an attractive sugar signaling molecule for Actinobacteria, especially *Streptomyces*[20]. According to our model, the strict response to GlcNAc occurs under the allosteric regulation of c-di-AMP. On the one hand, c-di-AMP binds DasR as a corepressor to control GlcNAc utilization, blocking the direct import of external GlcNAc-induced signaling. On the other hand, c-di-AMP binding to DasR enhances its positive effect on sporulation and antibiotic production. Thus, the cells relieved the repression caused by GlcNAc, faithfully maintaining a robust state for bacterial development and the production of secondary metabolites. In fact, GlcNAc catabolism is also involved in the regulation of virulence and pathogenesis of various human pathogens, including *Vibrio cholerae, Candida albicans*, and *Mycobacterium*, providing a survival advantage to pathogens in the host. Our proposed model involving GlcNAc sensing and signaling mechanisms could provide more targets to reduce the virulence of these severe pathogens. However, detailed examinations during infectious host–pathogen interactions are still scarce.

In all domains of life, nucleotide-based second messengers allow rapid integration of external and internal signals into regulatory pathways that control cellular responses to changing conditions. Previous studies have well explored that c-di-GMP acts as a brake to developmental transition through BldD and WhiG in the streptomyces life cycle[37–39]. Previous work from our laboratory showed that BldD is a validated target of DasR in *S. erythraea*[26], indicating a direct linkage between the regulatory effects caused by c-di-AMP and c-di-GMP. Based on our results here, c-di-AMP is an activator of the developmental transition through DasR, as our genetic studies revealed that O*disA* sporulates precociously accompanied by prominent antibiotic promotion, probably offering diverse and cooperative functions among different second messengers. This fine-tuning enables the organism to recruit an appropriate signaling molecule and to decide between growth and sporulation. In addition, the c-di-AMP-DasR interaction dominates the complete GlcNAc signaling cascade, conferring on it a vast range of biological roles from cell metabolism to signaling pathways. Our findings provide a coherent network that lays the foundation for deciphering c-di-AMP signaling towards a comprehensive understanding of global regulatory circuits that govern Actinobacteria states. We were also intrigued by the possibility that adjustment in the amount of intracellular c-di-AMP, in principle, could be a strategy for increased exploitation of the biotechnological potential of Actinobacteria. Thus, a detailed molecular understanding of the coordinated regulation of c-di-AMP and this crucial signaling network is of both fundamental and practical interest.

## Methods

### Bacterial strains and culture conditions
The strains and plasmids used in the experiment are listed in Supplementary Table 2. *E. coli* DH5α was used for plasmid propagation. *S. erythraea* strains were grown in tryptic soy broth (TSB) medium for 48 h at 30 °C for seed-stock preparation. Then, 0.5 mL of the seed cultures were inoculated in a 500-mL flask containing 50 ml TSB medium or supplemented with 45 mM GlcNAc for the indicated experiment. GlcNAc concentration was measured by the Morgan-Elson procedure[30]. Fifty microliters of extract were mixed with 75 μl 2 M potassium borate (12.36 g boric acid/100 ml, adjusted to pH 9.2 with KOH), boiled for 3 min, and then cooled to room temperature in water. Then 625 μl of Ehrlich's reagent (1 g 4-(dimethylamino) benzaldehyde acidified with 1.25 ml HCl and dissolved in 100 ml glacial acetic acid)

was added and incubated at 37 °C for 25 min. The tubes were centrifuged at 4 °C to cool them and remove any precipitated material, and the optical density of the purple color was measured at 585 nm. Standard curves were constructed using GlcNAc.

Phenotypic characterization of mutants was carried out on R2YE agar plates at 30 °C. Colonies were grown for 72 h on R2YE agar and prepared for scanning electron microscopy. Each specimen was examined with a Hitachi S-4800 (Japan) scanning electron microscope.

### Erythromycin determination
Erythromycin was extracted from cultures grown for 120 h in 50 ml TSB at 30 °C. The erythromycin concentration was measured using an Agilent 1100 HPLC System with a C18 column (5 μm, 250 × 4.6 mm), which was equilibrated with 45% solution A (K$_2$HPO$_4$, 30 mM, pH 8.0) and 55% solution B (acetonitrile). An isocratic program was carried out at a flow rate of 1 mL min$^{-1}$ using a UV detector at 215 nm.

### Overproduction and purification of proteins
All genes were amplified by PCR from the genomic DNA of *S. erythraea* using the primers shown in Supplementary Table 3. After the gene sequence was verified by sequencing (Tsingke Biotechnology Co., Ltd.), the recombinant plasmid was introduced into *E. coli* BL21 (DE3). The *E. coli* cells were grown in LB medium containing 1% kanamycin to an OD$_{600}$ of 0.6. Protein expression was induced with 0.5 mM isopropyl β-D-1-thiogalactopyranoside and then incubated at 20 °C for 12–14 h. For protein purification, cells were collected by centrifugation, resuspended in phosphate-buffered saline (PBS) buffer (137 mM NaCl, 2.7 mM KCl, 10 mM Na$_2$HPO$_4$, 1.8 mM KHPO$_4$), washed twice, and broken by an ultrasonic cell crusher. Cell debris and membrane fractions were removed by centrifugation (15 min, 8000 × *g*, 4 °C). The proteins were loaded onto a Ni-nitrilotriacetic acid-Sepharose column (Qiagen). After discarding the flow-through, 20 mL washing buffer (50 mM NaH$_2$PO$_4$, 300 mM NaCl, and 20 mM imidazole, pH 8.0) was used to wash the column, and the bound protein was eluted by using a linear gradient of 20–250 mM imidazole. The quality of the purified protein was determined by SDS polyacrylamide gel electrophoresis (SDS-PAGE). The protein concentration was determined by the bicinchoninic acid method, with bovine serum albumin as the standard.

### EMSA
The putative promoter region of the detection gene used for EMSA (−150 to +150 predicted to contain the DisA binding site) was amplified by PCR from *S. erythraea*, where the gene-specific primers contained universal primer sequences (5′-AGCCAGTGGCGATAAG-3′) and then labeled 5′-biotin with a universal primer (Supplementary Table 3). The PCR product was purified with an EMSA probe PCR purification kit (Tiangen Biotechnology, China), and its concentration was measured with a microplate reader (Bio Tek, USA). EMSAs were performed according to the protocol attached to the chemiluminescence EMSA kit (Beyotime Biotechnology, China). The binding reaction solution contained 10 mM Tris-HCl (pH 8.0), 25 mM MgCl$_2$, 50 mM NaCl, 1 mM DTT, 1 mM EDTA, 0.01% Nonidet P-40, 50 μg/mL poly(d[IC]), and 10% glycerol. The biotin-labeled DNA probe (1 ng) was incubated with different amounts of protein separately at 18 °C for 30 min. Then, samples were dissolved on a 6.5% nondenaturing PAGE gel in ice-cold 0.5× Tris-Borate-EDTA at a voltage of 100 V, and the bands were detected by an Omni-ECL™ Pico Light Chemiluminescence Kit (Epizyme Biotechnology, China).

### Real-time RT-PCR
The cells grown at the indicated time point were harvested by centrifugation (12,000 × *g*, 4 °C for 5 min), and an RNAprep Pure Cell/Bacteria kit (Tiangen Biotechnology, China) was used for RNA extraction. The RNA concentration was measured by a microplate

reader (Bio Tek, USA). Total RNA (1 μg) was reverse transcribed using the PrimeScript RT Reagent kit and gDNA Eraser (Takara). PCR was performed by using the CFX96 real-time system (Bio-Rad, USA). The following thermal cycling conditions were used: 95 °C for 5 min, then 95 °C for 15 s, 58 °C for 15 s, and 72 °C for 30 s for 40 cycles. The primers used are listed in Supplementary Table 3, and SACE_8101 was used as an internal control[40].

## c-di-AMP quantification

Bacteria were collected by centrifugation (8000 × $g$, 30 min), washed, and freeze-dried after centrifugation to determine the dry weight for standardization purposes. The precipitate was suspended in 15–20 mL of ice-cold cell extraction buffer (ether/methanol/$H_2O$ −40:40:20; LC-Grade, VWR rapid freezing for 5 min)[41]. The samples were instantly frozen in liquid $N_2$ for 10 min, heated to 95 °C for 10 min, and placed at 4 °C for 30 min. Then, a cell crushing machine was used to ultrasonically lyse for 15 min. The supernatant was removed and stored at 4 °C. The cell components were mixed and washed with extraction buffer, incubated on ice for 30 min and heated again. The sample was centrifuged again, and the supernatant was mixed with the previous supernatant. After freeze-drying, samples were resuspended in 500 μL cell extraction buffer for HPLC[42] with a chromatographic analysis column RPC-18 column (250 × 4.5 mm; Kromasil). The c-di-AMP concentrations of the samples were normalized to the dry cell weight. The intracellular concentrations of c-di-AMP were calculated using a cell volume of 7.14 mL/g dry cell weight, which was based on that of *E. coli*.

## Biolayer interferometry (BLI) assay

Two biosensors were used in this work: streptavidin (SA) and anti-Penta-HIS (HIS1K), purchased from ForteBio. The loading buffer (pH 8.0) contained 10 mM HEPES, 2 mM $MgCl_2$, 0.1 mM EDTA, and 200 mM KCl, and the running buffer contained an extra 10 μg/mL BSA and 0.02% Tween-20. The biotin-labeled DNA probe used was the same as the EMSAs. The DNA probe was stored in a loading buffer, and His-tagged DasR was stored in a running buffer during the BLI assay with SA sensors. When HIS1K sensors were used, the DasR protein was replaced with that stored in the loading buffer, and c-di-AMP was diluted with running buffer. Samples were then detected within the OptiPlate-96 Black Opaque (PerkinElmer).

## ITC experiment

Purified DasR protein was titrated with c-di-AMP in MicroCal iTC200. All samples were prepared in PBS buffer (pH 7.5). Typically, the titrant concentration in the syringe was 200–500 μM, and the titrant concentration in the reaction cell was 10–30 μM[43]. Titration was conducted at 25 °C using a multiple injection method with 150 s intervals. The obtained data were integrated, corrected, and analyzed using the MicroCal PEAQ-ITC Analysis Software with a single-site binding model.

## CD assay

Before and after incubation with c-di-AMP, samples were evaluated using CD spectrometry (Applied Photophysics, Leatherhead, UK) in the far-UV region (180–260 nm) at room temperature using a 10-mm cuvette. The proteins (0.2 mg/mL) were dissolved in a modified PBS buffer (pH 7.4) containing 1.4 M KF, 100 mM $K_2HPO_4$, and 18 mM $KH_2PO_4$. The CD spectrum scan of every sample was performed in triplicate. The spectra were analyzed for secondary structure content using CDNN CD spectra deconvolution software.

## Enzymatic assay

The bacteria were collected, resuspended, and washed with PBS. Then, a cell-crushing machine was used to lyse the cells ultrasonically and centrifuge them to obtain the whole protein. Equal amounts of protein (10 mg) from different strains were reacted with the reaction system at 37 °C for 2 h. The reaction system included 40 mM HEPES (pH 7.5),

0.2 mM ATP, 100 mM NaCl, 10 mM $MgCl_2$, and 10 mM protein. Afterward, the reaction was terminated at 95 °C for 5 min and centrifuged, and the supernatant was collected for c-di-AMP detection.

## ChIP-seq and data analysis

WT or O*disA* strains were grown in TSB liquid until late exponential growth (48 h). Formaldehyde was added to cultures at a final concentration of 1% (vol/vol), and incubation was continued for 30 min. Glycine was then added to a final concentration of 125 mM to stop the cross-linking. The samples were left at room temperature for 10 min and washed twice in precooled TBS buffer (pH 7.5) containing 20 mM Tris-HCl and 150 mM NaCl. ChIP-seq was performed by E-GENE Tech Co., Ltd. (Shenzhen) using an anti-DasR polyclonal antibody (1:200 dilution, Beyotime Biotechnology, China). Briefly, Fastp software (version 0.20.0) was used to trim adaptors and remove low-quality reads to obtain high-quality clean reads. Clean reads were aligned to the reference genome using Bowtie2 software (version 2.3.4.3). MACS2 software (version 2.1.2) was used for peak calling. Bedtools software (version 2.30.0) was used for peak annotation based on GTF annotation files. Homer (version 4.11) software was used to identify motifs. MAnorm2 (version 1.2.0) software was used to identify differentially enriched regions. The enriched peaks were visualized in IGV (version 2.4.10) software.

## Overexpression of DisA or DasR from *S. erythraea*

Using *S. erythraea* genomic DNA as a template, the *disA* or *dasR* gene was amplified by PCR. The vector pIB139[44] containing the *ermE* promoter was used for the expression of *disA*, and the *ermE* promoter is constitutive in *S. erythraea*. The vector pIB139 has been verified to have no effects on the bacterial physiology[45]. The PCR product was digested with EcoRV and NdeI and then inserted into the corresponding sites of the integrative plasmid pIB139. By PEG-mediated protoplast transformation, the overexpression strains were obtained by apramycin resistance screening and confirmed by PCR analysis with the primers apr-F and apr-R[40].

## Chemical cross-linking experiment

DasR proteins were dialyzed into the cross-linking buffer (100 mM $NaH_2PO_4$, 150 mM NaCl, pH 8.0) and then incubated at 25 °C for 1 h in reaction samples containing 10 mM protein, 2 mM disuccinimidyl suberate in dimethylsulfoxide, and c-di-AMP as indicated. Samples were fractionated on 15% SDS-PAGE gels and visualized by Coomassie blue staining. All experiments were performed at least twice.

## Molecular docking analysis

The three-dimensional (3D) structure of *S. erythraea* DasR (Supplementary Data 1) was predicted in Alphafold2. The structure of *S. coelicolor* DasR (PDB; 4ZS8[46]) was retrieved from PDB. The structure of c-di-AMP was retrieved from PubChem. Molecular docking was performed on *S. erythraea* DasR with c-di-AMP (Supplementary Data 2), and we assumed that the stoichiometric ratio was 1:1 (the most common binding stoichiometry). The SDF file format of c-di-AMP was converted to PDB format using Openbabel (version 2.4.1). The PDB file formats of c-di-AMP and DasR were converted to AutoDock PDBQT format using MGLTools (version 1.4.6). The removal of water molecules, the addition of hydrogen atoms, and charges in the receptor were considered prerequisites of predocking. Molecular docking studies were conducted using AutoDock (version 4.2.6). The docking center coordinates X, Y, and Z were set as follows: center_x = −5.907, center_y = −0.118, and center_z = 6.553. All dimensions of the docking box were divided into a grid of 122 × 84 × 110 points with a grid spacing of 0.375 Å. The maximum limit when searching for conformations was set to 2000, and the genetic algorithm was used to sample and score conformations via semiflexible docking. The best pose of c-di-AMP was determined based on binding energy (kcal/mol) to reflect the

outcomes of docking. The PyMOL (version 2.5.5) Molecular Graphics System was used to visualize H-bond interactions, binding affinities, interacting amino acid residues, atoms involved in ligand-receptor binding, and 3D graphical representations of ligand-receptor complexes.

## Reporting summary

Further information on research design is available in the Nature Portfolio Reporting Summary linked to this article.

## Data availability

Data representations can be found in the article and the Supplementary Information file. The molecular docking data generated in this study are provided in Supplementary Data 1 and 2. The ChIP-seq data used in this study are available in the GEO database (https://www.ncbi.nlm.nih.gov/geo/query/acc.cgi?acc=GSE229938). Source data are provided with this paper.

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

## Acknowledgements

This study was supported by grants from the National Key Research and Development Program of China (2018YFA0900404 to Bang-Ce Ye, 2020YFA0909100 to Di You), the National Natural Science Foundation of China (32070066 to Di You), the Fundamental Research Funds for the Central Universities (JKF01221617, JKF01231816 to Di You), and Shanghai Pilot Program for Basic Research (22TQ1400100-14 to Di You). We thank the staff members of the Large-scale Protein Preparation System at the National Facility for Protein Science in Shanghai (NFPS), Shanghai Advanced Research Institute, Chinese Academy of Sciences, China, for providing technical support and assistance in data collection and analysis.

## Author contributions

Di You designed research; Di You, Liu-Chang Zhao, Yu Fu, Zhi-Yao Peng, and Zong-Qin Chen performed research; Di You, Yu Fu, and Liu-Chang Zhao analyzed data; Bang-Ce Ye refined the ideas; and Di You and Bang-Ce Ye wrote the paper.

## Competing interests

The authors declare no competing interest.
