## [Peer Review File · Nature Communications]

Allosteric regulation by c-di-AMP modulates a complete N-acetylglucosamine signaling cascade in *Saccharopolyspora erythraea*

Editorial Note: Parts of this Peer Review File have been redacted as indicated to remove third party material where no permission to publish were obtained.Reviewers' comments:

Reviewer #1 (Remarks to the Author):

The manuscript by Di You and colleagues presents a variety of genetic data in support of the possible interaction of the actinobacterial GntR-family transcriptional regulator DasR with the second messenger c-di-AMP. The presented data look mostly convincing and are consistent with the hypothesis that c-di-AMP binding by DasR is a key part of the N-acetylglucosamine signaling pathway. If confirmed, this could be an important contribution to a better understanding of the multiple functions of c-di-AMP in actinobacteria. Having said that, I believe that the publication of these results needs to address the following points.

1. Fig. 1A presents only relative levels of c-di-AMP in the wild-type and OdisA strains. It would be essential to calculate actual intracellular concentrations of c-di-AMP. These numbers are necessary to show whether the K_d value of 26.8 μM c-di-AMP obtained by titrating its interaction with DasR using the ITC is physiologically relevant. Further, most experiments described in this paper use extremely high levels of c-di-AMP (100 μM , even 300 μM). It is essential to explain such a choice and state which effects can be seen at the likely physiological levels of c-di-AMP (i.e. 1-10 μM). Also, would DarS-c-di-AMP interaction be sensitive to the competition by other nucleotides (ATP, ADP, AMP, c-di-GMP)?

2. This work makes no attempt to characterize the putative c-di-AMP-DasR complex. It is quite strange given the availability of a high-resolution structure of DasR from *Streptomyces coelicolor* (PDB: 4ZSI) reported in 2016 by Fillenberg et al. (PubMedID: 27337024). The availability of a structure of this complex would resolve all questions regarding the c-di-AMP binding to DasR.

3. Transcriptional regulators of the GntR family are widespread in bacteria. It appears that only a small fraction of them could bind c-di-AMP. The suggestion that GntR from *E. coli* or *Salmonella enterica* could do that (lines 307-308) is hardly justified, given that *E. coli* cannot produce c-di-AMP (except for several pathogenic strains, recently reviewed by Galperin, PubMedID: 37022175). If the authors really believe that, they should perform the appropriate experiments.

4. The authors repeatedly state that "signal integration mediated through DasR is likely conserved and essential among Actinobacteria" (lines 117-118 and others). However, the NCBI's COG website <https://www.ncbi.nlm.nih.gov/research/cog/cog/COG1624/>

shows that more than 40% of actinobacterial genera do not encode diadenylate cyclases (see also <https://www.ebi.ac.uk/interpro/entry/pfam/PF02457/taxonomy/uniprot/#sunburst>). The paper would

greatly benefit from a comparison of those DasRs that are expected to bind c-di-AMP and those that are not.

5. The Introduction section and the legends to figures would need to be carefully revised by a professional speaker of English. Also, many references list only the volume but not the pages (or the entry codes). Some of the key errors in the text are listed below.

Line 57. Change 'then' to 'which'

Line 64-65. "expanding importance" – rephrase.

Line 65. "bacteria encounter" – rephrase.

Line 66. Change 'nutrients' to 'nutrient'

Line 67. "utilize mechanisms" -rephrase.

Line 78, and 79. Change 'contains' to 'containing'

Line 80. Remove "have been identified".

Line 86. Change "evidences suggest" to "evidence suggests".

Line 90. *Candida albicans* is NOT a bacterium.

Line 93. Many actinobacteria are not spore-forming.

Line 99. "this genus". Which genus?

Line 133, 140. "under GlcNAc conditions". What do you mean?

Fig. 2G. What is DSS?

Reviewer #2 (Remarks to the Author):

The control of antibiotic production and development in Actinobacteria is an important topic that is not yet well understood, despite many discoveries in the past half a century or so. The authors here show that the key signalling molecule cyclic-di-AMP controls DasR, which is a major finding that is a new example of how different primary metabolic and signalling routes are interconnected. The authors are commended for this discovery, and for the important work they have done on this exciting topic.

Having said that, there is a major issue to be addressed, and this is due to a controversy in the work, and to the fact that the authors have missed the importance of growth conditions on the activity of DasR. As shown in the Feast and Famine paper on DasR (Rigali et al., EMBO Reports 2008), DasR acts as a

repressor of antibiotic production and development on Minimal media (Famine), but acts as an activator under rich growth conditions (Feast). The authors have missed the fact that if c-di-AMP apparently activates binding of DasR and if indeed DasR is a repressor, you would expect hyper-repression! Instead, antibiotic production and development are activated and not blocked. Clearly, this must be resolved. My comments are below.

1. Throughout the paper, the authors mention that DasR is primarily a repressor. As shown in Rigali 2008, that is true for Minimal Media, but those were not the conditions analysed in the paper. Instead, all work was done in TSB (I think, see next point) or R2YE agar plates. Those are “Feast” conditions, and here antibiotic production and development is repressed by DasR, or at least adding GlcNAc blocks AB and Development. That is precisely what is seen in the R2YE plates in Figure 1, so *Sacch erythraea* and *S. coelicolor* are likely comparable. In the abstract it says “Strikingly, we show that increased c-di-AMP levels activate DasR as a repressor, causing the entire shutdown of the GlcNAc signaling cascade, leading to a consistent enhancement (!!) of development and antibiotic production in Actinobacteria”. See also the section lines 142-150. This part of the work is controversial. If DasR represses AB production, how then can activation of DasR by c-di-AMP lead to enhanced production of erythromycin? If c-di-AMP makes DasR bind stronger, which is convincingly shown in the paper, and this leads to superbinding, then repression should be complete. This is indeed the case for the nag genes in Fig. 3E. However, it appears that under the chosen conditions, DasR acts as an ACTIVATOR of AB and Development. That most likely means that you see opposite activities between R2YE and TSB grown cultures.

That really needs to be worked out as these observations are in contradiction and show that the system is much more complex than the authors claim. My question to the authors is therefore, can you give a plausible and experimentally validated explanation for what happens?

[a] What is missing in the first place is MM and TSB agar plates in Figure 1, to compare with the R2YE plates. Perhaps indeed you would see the opposite effect for TSB, i.e. TSB being similar to MM (Famine, with activated development and antibiotic production)?

[b] real time PCR is required for genes involved in development and antibiotic production, and certainly the ery cluster. If that also shows reduced expression, there is a major issue as ERY production is enhanced.

[c] a dasR null mutant would be extremely useful. How does the phenotype of del DasR compare to overproduction of c-di-AMP?

2. To address point 1, I would strongly recommend doing experiments in Minimal Media with a very poor carbon source. For *S. coelicolor* Agar was used as sole carbon source, but that will not be sufficient for *Saccharopolyspora* as it does not produce agarase. Mannitol may be an alternative, though it is already quite a good source of nutrients.

3. It is often not very clear under which conditions culturing was done. I am assuming all was done in TSB, except for the agar plates? See e.g. also line 213 “under the same conditions”, what are those

conditions? And line 408 “erythromycin was extracted from the 50 ml cultures for 120 h”. I expect cultures were not extracted for 120 h. Do the authors mean “... extracted from cultures grown for 120 h in 50 ml TSB at 300C”? Adding the media and mode of cultivation (liquid/solid, time of incubation) to the Figure legends and each time to the text would help a lot. Indicate among others media, volume and time of incubation (in hours).

4. In the section around line 307, the authors claim there are DasR orthologues in Gram-negatives. As far as I am aware, DasR is restricted to Gram-positives. In *Bacillus* or *Streptococcus*, DfasR controls GlcNAc metabolic genes and then a small set of other genes. In Actinobacteria, control is much more widespread. Every bacterium will have GntR regulators, how can the authors be sure that those in Gram negatives are DasR homologues? That would mean you should be able to find dre sequences upstream all GlcNAc-related genes, and need experimental evidence. I would remove the part on Gram-negatives, or add extensive argumentation including experiments why these are considered true orthologues.

5. Major factors that control antibiotic production and development are Fe and phosphate. DasR controls siderophore biosynthesis and thus Fe homeostasis, which makes R2YE a special media because it is deprived of iron. As far as I know, TSB stands out as being rich in phosphate. Again, it would be useful to look at MM cultures.

6. In Figure 1, it is shown that cAMP does not bind DasR. However, in Fig. 5, cAMP has a small effect. I would strongly recommend performing ITC also on cAMP and preferably cGMP and c-di-GMP as controls.

7. I missed the link with c-di-GMP control (Buttner lab). This runs via an other global regulator, BldD. Please consider adding to the paper. Do the authors see a link between the two systems?

8. Line 111. I suppose breakdown means metabolism? The whole section needs to be modified, as DasR represses GlcNAc metabolism and does not mediate its ‘breakdown’, more the opposite. It prevents uptake and utilization of aminosugars.

Reviewer #3 (Remarks to the Author):

This manuscript by You et. al. reports a novel receptor of the second messenger, cyclic di-AMP (cdA) in Actinobacteria. The authors generate a cdA overexpression strain, and observe differences in the utilization of N-acetylglucosamine (GlcNac), sporulation, and antibiotic production. The authors conclude that these differences are mediated by the central transcription factor, DasR, which was previously shown to bind and be inhibited by GlcNac. cdA, on the other hand, enhances the repression activity of DasR. Finally, DasR represses the cdA synthase, *disA*, suggesting that cdA negatively feedbacks on its own production. In sum, this is an interesting finding that expands what we know about cdA signaling. My biggest concern is the K_d of DasR binding to cdA was very high and likely not a physiologically relevant concentration of cdA and the concentrations of cdA used in the in vitro experiments, like the EMSAs, were quite high. Without showing the effects of cdA at more physiological concentrations, I have some doubts as to the validity of the model. This and other specific suggestions are listed below:

1. Line 57, change “then” to “which”
2. Line 64-66-This is not a complete sentence. The manuscript is well written, but it could use some English editing to tighten up a few spots like this.
3. It would also be nice in the discussion if the authors could integrate their identification of DasR as a new receptor of c-di-AMP with the others that have been discovered. Specifically, is this an entirely new class or a new example of an existing class?
4. Fig. 1C-the authors need to specify which line is associated with which axis (solid for growth and dashed for utilization) in the figure legend.
5. Line 122-the authors should give some details here about how *disA* was overexpressed. What was the promoter used? Was an inducer added? Was this on a plasmid? Etc. And if a plasmid was used, was a vector control with inducer also present in the WT strain to control for non-specific effects of these components?
6. Figure 1 legend-The authors should specify in this legend and other what statistical analysis was used to determine statistical significance.
7. Fig. 2C-A K_d of 26.8 uM seems quite high as most cyclic di-nucleotides are present in low uM to nM concentrations in the cell. What are the physiological concentrations of cdA in *Streptomyces*?
8. Fig. 2D-cdA impacts shifting of DasR, but 100 uM is used, which is a very high amount that is not physiologically relevant. In order to support their conclusion, the authors should demonstrate this effect at lower concentration of cdA. Another way to strengthen these data is to demonstrate this effect is specific for cdA and such enhancement of DNA binding does not occur with other cdNs like c-di-GMP or cGAMP.
9. Fig. 2G-What concentrations of cdA are used here?
10. Fig. 2G-the protein ladder should be indicated to show the size of the monomers and dimers.

11. Fig. 3E-A nice control for this experiment would have been to demonstrate that the cdA repression of these genes does not occur in a dasR mutant. In other words, analysis of a dasR versus a dasR/OdisA should not demonstrate cdA repression.

12. Fig. 3b-I am confused by what this represents. Is this the summary of all ChIP-seq signals or a representative peak?

13. Fig. 4C-The authors use a dasR deletion strain, but in lines 118 and 302 they state that this gene is essential, which means that you cannot generate a null mutation and still have viability. Could the authors clarify.

14. Lines 353-356-What the authors are describing here is really a negative feedback loop on cdA synthesis to limit overproduction of this signal.

15. Fig. 6-One central question that I was left with and was unaddressed is how cdA through DasR leads to enhanced sporulation and antibiotic production. Is DasR not functioning as a repressor for these genes or is there an intermediate regulator. Does DasR directly bind to the promoters of these genes in the ChIP-seq experiment. Some discussion around lines 371-372 is warranted.

Response to the reviewers

Reviewers' comments:

Reviewer: 1 (Remarks to the Author):

The manuscript by Di You and colleagues presents a variety of genetic data in support of the possible interaction of the actinobacterial GntR-family transcriptional regulator DasR with the second messenger c-di-AMP. The presented data look mostly convincing and are consistent with the hypothesis that c-di-AMP binding by DasR is a key part of the N-acetylglucosamine signaling pathway. If confirmed, this could be an important contribution to a better understanding of the multiple functions of c-di-AMP in actinobacteria. Having said that, I believe that the publication of these results needs to address the following points.

Author response:

We greatly appreciate the reviewer's positive comments, as well as the time and the effort spent reviewing our work. The manuscript has been carefully revised in accordance with the reviewer's suggestion. The necessary experiments and missing information were also added, which we hope will be met with approval.

- (1) Fig. 1A presents only relative levels of c-di-AMP in the wild-type and OdisA strains. It would be essential to calculate actual intracellular concentrations of c-di-AMP. These numbers are necessary to show whether the K_d value of 26.8 μM c-di-AMP obtained by titrating its interaction with DasR using the ITC is physiologically relevant. Further, most experiments described in this paper use extremely high levels of c-di-AMP (100 μM , even 300 μM). It is essential to explain such a choice and state which effects can be seen at the likely physiological levels of c-di-AMP (i.e. 1-10 μM). Also, would DarR-c-di-AMP interaction be sensitive to the competition by other nucleotides (ATP, ADP, AMP, c-di-GMP)?

Author response:

Thank you for the comments. We have changed the relative levels of c-di-AMP to absolute quantification, as shown in revised Fig. 1A. The c-di-AMP concentrations of the samples were normalized to the dry cell weight. The intracellular concentrations of c-di-AMP were calculated using a cell volume of 7.14 mL/g dry cell weight, which was based on that of *E.coli*. The estimated c-di-AMP concentration in *S. erythraea* WT strain is $\sim 40 \mu\text{M}$. Moreover, in strains in which the c-di-AMP level was artificially increased 2~5-fold ($\Delta dasR$ or *OdisA*), these numbers confirmed that c-di-AMP binding could occur under physiological conditions. Previous studies have also reported that intracellular concentrations of c-di-AMP are approximately 0.5–70 μM ^[1-3], including 28 μM in *Synechococcus elongatus* ^[1] and 70 μM in *Staphylococcus aureus* ^[2], further supporting that c-di-AMP binding by DasR should be relevant *in vivo*.

Figure 1. (A) *disA* transcription levels and intracellular c-di-AMP concentrations in WT and *OdisA* strains grown in liquid TSB medium during late exponential growth (48 h). Fold change represents the expression level or c-di-AMP level compared to the WT strain. The c-di-AMP concentrations of the samples were normalized to the dry cell weight.

The high levels of c-di-AMP (100 μ M for EMSA, 300 μ M for ITC) used in the study were optimized to perform all assays under substrate-saturating conditions according to the experimental principles [4]. ITC experiments are optimally performed to obtain a full description of the interaction (e.g., Kd 10 μ M would require 30 μ M in the cell and thus at least 300 μ M of ligand in the syringe) [4]. In the case of EMSA, we demonstrated this effect with lower concentrations of c-di-AMP. As shown in Figure S3A, the DNA-binding activity of DasR was enhanced with progressively increasing amounts of c-di-AMP, and further analysis using a relevant concentration of c-di-AMP (30 μ M) confirmed that the enhancement of DNA binding occurred with lower concentrations of c-di-AMP (Figure S3B). These figures were added to the **Supporting Information** section.

Figure S3. EMSA of DasR binding to its target gene *nagA* promoter. (A) Purified His-DasR (15 μ M) and a 300-bp DNA fragment were incubated with a gradient concentration of c-di-AMP (1, 5, 10, 30, 100, and 120 μ M). (B) Purified His-DasR and a 300-bp DNA fragment were incubated without c-di-AMP (left) or with 30 μ M c-di-AMP (right).

According to the reviewer's comments, we performed ITC assays using DasR titrated with ATP, ADP, AMP, and c-di-GMP. The results showed that none of the detected nucleotides could bind DasR (Figure S2). These observations support a specific interaction with c-di-AMP. These

figures were added to the Supporting Information section.

Figure S2. Characterization of the interaction of nucleotides with DasR using ITC. Titration of nucleotides (300 μM) into DasR (30 μM). ITC measurement of DasR titrated with ADP (A), AMP (B), ATP (C), and c-di-GMP (D).

- [1] Gomelsky M, Rubin BE, Huynh TN, Welkie DG, Diamond S, Simkovsky R, Pierce EC, Taton A, Lowe LC, Lee JJ, Rifkin SA, Woodward JJ, Golden SS. 2018. High-throughput interaction screens illuminate the role of c-di-AMP in cyanobacterial nighttime survival. *PLoS Genet* 14:e1007301.
- [2] Rebecca, M., Corrigan, Lisa, Bowman, Alexandra, R., Willis, Volkhard, Kaever. 2015. Cross-talk between Two Nucleotide-signaling Pathways in *Staphylococcus aureus*. *J Biol Chem* 290:5826-5839.
- [3] He J, Yin W, Galperin MY, Chou SH. 2020. Cyclic di-AMP, a second messenger of primary importance: tertiary structures and binding mechanisms. *Nucleic Acids Res* 48:2807-2829.
- [4] Williams MA, Daviter T. 2013. *Protein-Ligand Interactions: Methods and Applications*. *Methods Mol Biol*.

- (2) This work makes no attempt to characterize the putative c-di-AMP-DasR complex. It is quite strange given the availability of a high-resolution structure of DasR from *Streptomyces coelicolor* (PDB: 4ZSI) reported in 2016 by Fillenberg et al. (PubMedID: 27337024). The availability of a structure of this complex would resolve all questions regarding the c-di-AMP binding to DasR.

Author response:

Thank you for the very helpful suggestion. We used AlphaFold2 to predict the possible structure of DasR. DasR from *Streptomyces coelicolor* (PDB:4ZSI^[5]) served as a template for the structural model. AutoDock was used for the prediction of c-di-AMP binding. Molecular docking indicated that binding of c-di-AMP occurred in the effector-binding domain of DasR, and Arg246, Asn248, Leu91 and Gln92 might contribute to the interaction with c-di-AMP via hydrogen bonding (Figure S4). These figures and descriptions were added to the revised manuscript. We will also characterize the actual binding mode of c-di-AMP in DasR for our further study in this field.

Figure S4. The putative c-di-AMP-DasR complex. (A) Model of DasR obtained from Alphafold2 (blue) superimposed with 4ZSI (gray). The close-up shows the predicted interaction sites, annotated with the putative binding residues. (B) A zoomed view of interactions between c-di-AMP and DasR. (C) A zoomed view of interactions between c-di-AMP and 4ZSI. Interactions are denoted with labels.

[5] Fillenberg SB, Friess MD, Korner S, Bockmann RA, Muller YA. 2016. Crystal Structures of the Global Regulator DasR from *Streptomyces coelicolor*: Implications for the Allosteric Regulation of GntR/HutC Repressors. *PLoS One* 11:e0157691.

- (3) Transcriptional regulators of the GntR family are widespread in bacteria. It appears that only a small fraction of them could bind c-di-AMP. The suggestion that GntR from *E. coli* or *Salmonella enterica* could do that (lines 307-308) is hardly justified, given that *E. coli* cannot produce c-di-AMP (except for several pathogenic strains, recently reviewed by Galperin, PubMedID: 37022175). If the authors really believe that, they should perform the appropriate experiments.

Author response:

Thank you for the very helpful comments. We have removed the statement on Gram-negatives and revised this section as “...To complement these evolutionary insights, the *S. erythraea* DasR sequence was aligned with homologs in other c-di-AMP-producing species, including SCO5231 from *S. coelicolor*, MSMEG_0286 from *Mycobacterium smegmatis*, and RV0792c from *Mycobacterium tuberculosis*, etc. (Figure 5A). Phylogenetic survey revealed that DasR homologs are broadly distributed among c-di-AMP synthesizing species, particularly for *Streptomyces* species. We therefore speculated that DasR as a c-di-AMP receptor would be comparatively conserved in streptomycetes and some other genera of actinobacteria....”.

Figure 5. (A) Phylogenetic analysis of the DasR protein in prokaryotes. Sequences of DasR proteins were aligned with MAFFT v7.489 with L-INS-I. Trees were reconstructed from the resulting alignments using FastTree and subsequently visualized interactive Tree of Life (iTol) Version 6.5.6. The green bar represents the aa length of the DasR protein. Left panels show the genus names analyzed in the phylogenetic survey and the identity of the *S. erythraea* DasR sequence aligned with the corresponding homologs.

(4) The authors repeatedly state that “signal integration mediated through DasR is likely conserved and essential among Actinobacteria” (lines 117-118 and others). However, the NCBI’s COG website <https://www.ncbi.nlm.nih.gov/research/cog/cog/COG1624/> shows that more than 40% of actinobacterial genera do not encode diadenylate cyclases (see also <https://www.ebi.ac.uk/interpro/entry/pfam/PF02457/taxonomy/uniprot/#sunburst>). The paper would greatly benefit from a comparison of those DasRs that are expected to bind c-di-AMP and those that are not.

Author response:

Thank you for the nice suggestion. We have revised the mentioned statements as “... among c-di-AMP-producing Actinobacteria.” and added related discussion as “Most bacteria encode only one or a few DACs (1), and c-di-AMP is essential in many but not all bacterial species. The essentiality of c-di-AMP was dependent on the growth conditions (6). Thus, as a whole, the interdependencies between c-di-AMP and DasR conceivably require a high degree of coordination and involve numerous different cellular processes.”.

(5) The Introduction section and the legends to figures would need to be carefully revised by a professional speaker of English. Also, many references list only the volume but not the pages (or the entry codes). Some of the key errors in the text are listed below.

Author response:

We apologize for the mentioned inconvenience of the English language. According to the reviewer’s helpful suggestion, we have corrected the grammatical mistakes and carefully

revised the whole manuscript with the help of a native English speaker. The missing information was also added to the **References** list.

(6) Line 57. Change 'then' to 'which'

Author response:

Thank you for the comment. We have corrected it.

(7) Line 64-65. "expanding importance" – rephrase.

Author response:

Thank you for the comment. We have rephrased it as "Given the importance it plays...".

(8) Line 65. "bacteria encounter" – rephrase.

Author response:

Thank you for the comment. We have rephrased it as "diverse bacterial physiology...".

(9) Line 66. Change 'nutrients' to 'nutrient'

Author response:

Thank you for the comment. We have corrected it.

(10) Line 67. "utilize mechanisms' -rephrase.

Author response:

Thank you for the comment. We have rephrased it as "Components of the c-di-AMP signaling system have fundamental requirements to...".

(11) Line 78, and 79. Change 'contains' to 'containing'

Author response:

Thank you for the comment. We have corrected it.

(12) Line 80. Remove "have been identified".

Author response:

Thank you for the comment. We have deleted it.

(13) Line 86. Change "evidences suggest" to "evidence suggests".

Author response:

Thank you for the comment. We have corrected it.

(14) Line 90. *Candida albicans* is NOT a bacterium.

Author response:

We apologize for the mistake. We have corrected it as "pathogenic fungi such as *Candida albicans*".

(15) Line 93. Many actinobacteria are not spore-forming.

Author response:

Thank you for the comment. We have deleted “spore-forming”.

(16) Line 99. “this genus”. Which genus?

Author response:

We apologize for this confusion. We have corrected it as “antibiotic-producing soil Actinobacteria”.

(17) Line 133, 140. “under GlcNAc conditions”. What do you mean?

Author response:

We apologize for this confusion. We have clarified it as “in presence of GlcNAc”.

(18) Fig. 2G. What is DSS?

Author response:

Thank you for the question. It is disuccinimidyl suberate and we have described it in the **Methods** section.

Reviewer: 2 (Remarks to the Author):

The control of antibiotic production and development in Actinobacteria is an important topic that is not yet well understood, despite many discoveries in the past half a century or so. The authors here show that the key signalling molecule cyclic-di-AMP controls DasR, which is a major finding that is a new example of how different primary metabolic and signalling routes are interconnected. The authors are commended for this discovery, and for the important work they have done on this exciting topic.

Having said that, there is a major issue to be addressed, and this is due to a controversy in the work, and to the fact that the authors have missed the importance of growth conditions on the activity of DasR. As shown in the Feast and Famine paper on DasR (Rigali et al., EMBO Reports 2008), DasR acts as a repressor of antibiotic production and development on Minimal media (Famine), but acts as an activator under rich growth conditions (Feast). The authors have missed the fact that if c-di-AMP apparently activates binding of DasR and if indeed DasR is a repressor, you would expect hyper-repression! Instead, antibiotic production and development are activated and not blocked. Clearly, this must be resolved. My comments are below.

Author response:

We greatly appreciate the reviewer's positive comments, as well as the time and the effort spent reviewing our work. These comments are valuable and very helpful for revising and improving our paper.

We agree that growth conditions are important for antibiotic production and development in Actinobacteria. Indeed, *dasR* absence leads to deficient in aerial hyphae and spore formation [6]. DasR has also been shown to repress GlcNAc utilization genes and secondary metabolite production in *Streptomyces* [7,8]. The excellent findings in the Feast and Famine paper further revealed that **GlcNAc** triggers or inhibits sporulation and secondary metabolite production in *Streptomyces* depending on the culture conditions [8].

Our previous study [9,10] has well established the common and distinct features of GlcNAc utilization and its transcriptional control between the erythromycin producer *Sacch. erythraea* and *Strep. coelicolor*. **The common feature** is that the expression of GlcNAc utilization genes is directly repressed by DasR, and DasR is an activator of development [9,10]. **A major difference** compared to the situation described in *Strep. coelicolor* is that DasR **stimulates** antibiotic production in *Sacch. erythraea* [10]. Hence, c-di-AMP binding activates DasR-mediated developmental transition and antibiotic production.

According to the reviewer's helpful suggestions, we have carefully revised the manuscript and added the necessary illustrations to avoid confusion.

[6] Seo, J. W., Ohnishi, Y., Hirata, A. & Horinouchi, S. 2002. ATPbinding cassette transport system involved in regulation of morphological differentiation in response to glucose in *Streptomyces griseus*. J Bacteriol 184, 91–103.

[7] Rigali S, Nothaft H, Noens EEE, Schlicht M, Colson S, Muller M, Joris B, Koerten HK, Hopwood DA, Titgemeyer F, van Wezel GP. 2006. The sugar phosphotransferase system of *Streptomyces coelicolor* is regulated by the GntR-family regulator DasR and links N-acetylglucosamine metabolism to the control of development. Mol Microbiol 61:1237-1251.

[8] Rigali S, Titgemeyer F, Barends S, Mulder S, Thomae AW, Hopwood DA, van Wezel GP. 2008. Feast or famine: the global regulator DasR links nutrient stress to antibiotic production by

Streptomyces. *Embo Reports* 9:670-675.

[9] Liao CH, Rigali S, Cassani CL, Marcellin E, Nielsen LK, Ye BC. 2014. Control of chitin and N-acetylglucosamine utilization in *Saccharopolyspora erythraea*. *Microbiology-Sgm* 160:1914-1928.

[10] Liao CH, Xu Y, Rigali S, Ye BC. 2015. DasR is a pleiotropic regulator required for antibiotic production, pigment biosynthesis, and morphological development in *Saccharopolyspora erythraea*. *Appl Microbiol Biotechnol* 99:10215-10224.

(1) Throughout the paper, the authors mention that DasR is primarily a repressor. As shown in Rigali 2008, that is true for Minimal Media, but those were not the conditions analysed in the paper. Instead, all work was done in TSB (I think, see next point) or R2YE agar plates. Those are “Feast” conditions, and here antibiotic production and development is repressed by DasR, or at least adding GlcNAc blocks AB and Development. That is precisely what is seen in the R2YE plates in Figure 1, so *Sacch erythraea* and *S. coelicolor* are likely comparable. In the abstract it says “Strikingly, we show that increased c-di-AMP levels activate DasR as a repressor, causing the entire shutdown of the GlcNAc signaling cascade, leading to a consistent enhancement (!) of development and antibiotic production in Actinobacteria”. See also the section lines 142-150. This part of the work is controversial. If DasR represses AB production, how then can activation of DasR by c-di-AMP lead to enhanced production of erythromycin? If c-di-AMP makes DasR bind stronger, which is convincingly shown in the paper, and this leads to super binding, then repression should be complete. This is indeed the case for the *nag* genes in Fig. 3E. However, it appears that under the chosen conditions, DasR acts as an ACTIVATOR of AB and Development. That most likely means that you see opposite activities between R2YE and TSB grown cultures.

That really needs to be worked out as these observations are in contradiction and show that the system is much more complex than the authors claim. My question to the authors is therefore, can you give a plausible and experimentally validated explanation for what happens?

Author response:

Thank you for the very helpful comments. Previous studies from Rigali et al. [7,8] and us [9,10] have all confirmed that GlcNAc blocks antibiotic production and development through DasR in both *Strep. coelicolor* and *Sacch. erythraea* under nutrient-rich conditions. This is one reason of why TSB and R2YE agar plates were used in this work. Another main reason is that the essentiality of c-di-AMP was dependent on the growth conditions, c-di-AMP is not essential in defined minimal media or rich media with lower salt concentrations [11-13]. Since this study focused on the interaction between c-di-AMP and GlcNAc signaling, we chose the nutrient-rich conditions in which both c-di-AMP and GlcNAc play biologically important roles in this study. We have added this description to the revised manuscript.

As we have just explained above, DasR stimulates antibiotic production in *Sacch. erythraea*, which is contrary to the situation described in *Strep. coelicolor*. In sum, DasR is indeed a pleiotropic regulator since it is a repressor of GlcNAc metabolism and an activator of antibiotic production and morphological development in *Sacch. erythraea*. Hence, c-di-AMP binding causes hyper-repression of GlcNAc metabolism and activates DasR-mediated developmental transition and antibiotic production. This is indeed the case for hyper-repression on both GlcNAc utilization (Fig. 1 C, samples cultured in TSB liquid) and the expression of *nag* genes

(Fig. 3 E, samples cultured in TSB liquid), and the activation of developmental transition (Fig. 1 D-G, samples cultured on R2YE agar) and antibiotic production (Fig. 1 H, samples cultured in TSB liquid). The conclusions at section lines 142-150 is hence reasonable.

We apologize for the confusion caused by “DasR is primarily a repressor”, and we have defined it as “DasR is primarily a repressor of GlcNAc utilization”. Accordingly, the mentioned sentence in abstract was revised as “Strikingly, we show that increased c-di-AMP levels activate DasR as a repressor of GlcNAc utilization, causing the entire shutdown of the GlcNAc signaling cascade, leading to a consistent enhancement of development and antibiotic production in *Saccharopolyspora erythraea*”.

[11] Davies BW, Bogard RW, Young TS, Mekalanos JJ. 2012. Coordinated regulation of accessory genetic elements produces cyclic dinucleotides for *V. cholerae* virulence. *Cell* 149:358-370.

[12] Pham HT, Nhiep NTH, Vu TNM, Huynh TN, Zhu Y, Huynh ALD, Chakraborti A, Marcellin E, Lo R, Howard CB et al. 2018. Enhanced uptake of potassium or glycine betaine or export of cyclic-di-AMP restores osmoresistance in a high cyclic-di-AMP *Lactococcus lactis* mutant. *PLoS Genet* 14:e1007574.

[13] Yoon SH, Waters CM. 2021. The ever-expanding world of bacterial cyclic oligonucleotide second messengers. *Curr Opin Microbiol* 60:96-103.

[a] What is missing in the first place is MM and TSB agar plates in Figure 1, to compare with the R2YE plates. Perhaps indeed you would see the opposite effect for TSB, i.e. TSB being similar to MM (Famine, with activated development and antibiotic production)?

Author response:

Thank you for the comments. Tryptic soy broth (TSB) and R2YE agar are nutrient-rich conditions. As TSB agar is known unsuitable for spore characterization, we analyzed the phenotype of WT and *OdisA* in MM medium. We found that *Saccharopolyspora erythraea* grew too weakly in either liquid MM or MM agar plates with glucose or GlcNAc as the carbon source (Figure R1), and it is almost impossible to analyze development and antibiotic production.

Figure R1. (A) Growth curves of *S. erythraea* WT, *OdisA* and $\Delta dasR$ strains grown at 30 °C in liquid TSB medium, liquid MM with glucose or GlcNAc as the carbon resource. (B) Phenotypes of *S. erythraea* WT, *OdisA* and $\Delta dasR$ strains grown at 30 °C on R2YE agar plates with glucose

addition, and MM agar with glucose or GlcNAc as the carbon resource.

[b] real time PCR is required for genes involved in development and antibiotic production, and certainly the ery cluster. If that also shows induced expression, there is a major issue as ERY production is enhanced.

Author response:

Thank you for the helpful comments. We have added the RT-PCR data of genes critical for development and antibiotic production including *bldD* and *ery* clusters. BldD is a key developmental regulator required for erythromycin biosynthesis [14,15], whose transcription is under direct activation of DasR [10]. As shown in Figure S1, *bldD* and *ery* clusters exhibited induced expression in *OdisA*, further confirming that erythromycin production was enhanced, and these figures were added to the **Supporting Information** section.

Figure S1. The transcription levels of the known genes critical for development and antibiotic production in *S. erythraea*. *S. erythraea* WT and *OdisA* strains were grown till late exponential growth (48 h) in liquid TSB medium with glucose (A) or GlcNAc (B) addition. Fold change represents the expression level compared to the WT strain. Error bars show the SDs of three independent experiments. Asterisks indicate a T-test significance value. *P < 0.05, **P < 0.01, ***P < 0.001, ****P < 0.0001.

[14] Chng,C., Lum,A.M., Vroom,J.A. and Kao,C.M. 2008. A key developmental regulator controls the synthesis of the antibiotic erythromycin in *Saccharopolyspora erythraea*. *Proc. Natl. Acad. Sci. U.S.A.*, 105, 11346–11351.

[15] Fu Y, Dong YQ, Shen JL, Yin BC, Ye BC, You D. 2023. A meet-up of acetyl phosphate and c-di-GMP modulates BldD activity for development and antibiotic production. *Nucleic Acids Res* 51:6870-6882.

[c] a *dasR* null mutant would be extremely useful. How does the phenotype of Δ *DasR* compare to overproduction of c-di-AMP?

Author response:

Thank you for the comments. The phenotype of $\Delta dasR$ was already investigated in our previous work ^[10], disruption of *dasR* causes delayed morphological defect, and GlcNAc was shown to repress the development of *S. erythraea* through DasR in the nutrient-rich conditions. We also performed a direct comparison using WT, $\Delta dasR$ and *OdisA* strains. As shown in Figure R2, $\Delta dasR$ strain exhibited an opposite character in development compared to *OdisA* strain in glucose, and shows comparable behavior to that of *OdisA* strain in GlcNAc.

Figure R2. The phenotype of $\Delta dasR$ and *OdisA* strains on nutrient-rich R2YE agar plates with glucose or GlcNAc addition.

[10] Liao CH, Xu Y, Rigali S, Ye BC. 2015. DasR is a pleiotropic regulator required for antibiotic production, pigment biosynthesis, and morphological development in *Saccharopolyspora erythraea*. *Appl Microbiol Biotechnol* 99:10215-10224.

- (2) To address point 1, I would strongly recommend doing experiments in Minimal Media with a very poor carbon source. For *S. coelicolor* Agar was used as sole carbon source, but that will not be sufficient for *Saccharopolyspora* as it does not produce agarase. Mannitol may be an alternative, though it is already quite a good source of nutrients.

Author response:

Thank you for the comments. In fact, the three strains of *S. erythraea* (WT, *OdisA* and $\Delta dasR$) showed similar growth restriction in MM medium with mannitol as carbon source (Figure R3), and further analysis was almost impossible.

Figure R3. Growth curves of *S. erythraea* WT, *OdisA* and Δ *dasR* strains grown at 30 °C in liquid TSB medium, liquid MM with mannitol as the carbon resource. Phenotypes of *S. erythraea* WT, *OdisA* and Δ *dasR* strains grown at 30 °C on MM agar with mannitol as carbon resource for 168 h.

- (3) It is often not very clear under which conditions culturing was done. I am assuming all was done in TSB, except for the agar plates? See e.g. also line 213 “under the same conditions”, what are those conditions? And line 408 “erythromycin was extracted from the 50 ml cultures for 120 h”. I expect cultures were not extracted for 120 h. Do the authors mean “... extracted from cultures grown for 120 h in 50 ml TSB at 30°C”? Adding the media and mode of cultivation (liquid/solid, time of incubation) to the Figure legends and each time to the text would help a lot. Indicate among others media, volume and time of incubation (in hours).

Author response:

Thank you for the questions. Yes, in the current study, we used nutrient-rich conditions (R2YE agar or TSB liquid) for practical purposes.

According to the reviewer’s helpful comments, line 213 “under the same conditions” was modified as “in TSB liquid”; line 408 “erythromycin was extracted from the 50 ml cultures for 120 h” was revised as “Erythromycin was extracted from cultures grown for 120 h in 50 ml TSB at 30 °C”. The mentioned missing details were all added to the revised manuscript.

- (4) In the section around line 307, the authors claim there are DasR orthologues in Gram-negatives. As far as I am aware, DasR is restricted to Gram-positives. In *Bacillus* or *Streptococcus*, DfasR controls GlcNAc metabolic genes and then a small set of other genes. In Actinobacteria, control is much more widespread. Every bacterium will have GntR regulators, how can the authors be sure that those in Gram negatives are DasR homologues? That would mean you should be able to find dre sequences upstream all GlcNAc-related genes, and need experimental evidence. I would remove the part on Gram-negatives, or add extensive argumentation including experiments why these are considered true orthologues.

Author response:

Thank you for the very helpful comments. We have removed the statement on Gram-

negatives and revised this section as “To complement these evolutionary insights, the *S. erythraea* DasR sequence was aligned with homologs in other c-di-AMP-producing species, including SCO5231 from *S. coelicolor*, MSMEG_0286 from *Mycobacterium smegmatis*, and RV0792c from *Mycobacterium tuberculosis*, etc. (Figure 5A). Phylogenetic survey revealed that DasR homologs are broadly distributed among c-di-AMP synthesizing species, particularly for *Streptomyces* species. We therefore speculated that DasR as a c-di-AMP receptor would be comparatively conserved in streptomycetes and some other genera of actinobacteria.”.

Figure 5. (A) Phylogenetic analysis of the DasR protein in prokaryotes. Sequences of the DasR proteins were aligned with MAFFT v7.489 with L-INS-I. Trees were reconstructed from the resulting alignments using FastTree and subsequently visualized interactive Tree of Life (iTol) Version 6.5.6. The green bar represents the aa length of the DasR protein. Left panels show the genus names analyzed in the phylogenetic survey and the identity of the *S. erythraea* DasR sequence aligned with the corresponding homologs.

- (5) Major factors that control antibiotic production and development are Fe and phosphate. DasR controls siderophore biosynthesis and thus Fe homeostasis, which makes R2YE a special media because it is deprived of iron. As far as I know, TSB stands out as being rich in phosphate. Again, it would be useful to look at MM cultures.

Author response:

Thank you for the comments. The reasons for the TSB and R2YE agar used in the current study is that nutrient-rich conditions provide an environment in which both c-di-AMP and GlcNAc play biologically important roles. Moreover, the results in Figure R2 and 3 showed growth restriction in MM medium, which is indeed impossible for further analysis.

- (6) In Figure 2, it is shown that cAMP does not bind DasR. However, in Fig. 5, cAMP has a small effect. I would strongly recommend performing ITC also on cAMP and preferably cGMP and c-di-GMP as controls.

Author response:

Thank you for the comments. Figure 2 showed the interaction of nucleotides with DasR from *Sacch. erythraea*, while Figure 5 indicated interactions with DasR from *Strep. coelicolor*. According to the reviewer's comments, we performed ITC assays using *S. erythraea* DasR titrated with cAMP, c-di-GMP and cGMP. The results confirmed that none of them binds DasR. We have added these data to the **Supporting Information** section.

Figure S2. Characterization of the interaction of nucleotides with DasR using ITC. Titration of nucleotides (300 μ M) into DasR (30 μ M). ITC measurement of DasR titrated with cAMP (A), c-di-GMP (B), and cGMP (C).

- (7) I missed the link with c-di-GMP control (Buttner lab). This runs via another global regulator, BldD. Please consider adding to the paper. Do the authors see a link between the two systems?

Author response:

Thank you for the nice suggestion. Yes, according to our findings in *S. erythraea*, BldD is a validated target of DasR^[10], indicating a direct linkage between the regulatory effects caused by c-di-AMP and c-di-GMP. We have added it to the **Discussion** section as “Previous studies have well explored that c-di-GMP acts as a brake to developmental transition through BldD and WhiG in the *Streptomyces* life cycle (39-41). Previous work from our laboratory showed that BldD is a validated target of DasR in *S. erythraea* (28), indicating a direct linkage between the regulatory effects caused by c-di-AMP and c-di-GMP. Based on our results here, c-di-AMP is an activator of the developmental transition through DasR, as our genetic studies revealed that *OdisA* sporulates precociously accompanied by a prominent antibiotic promotion, probably offering diverse and cooperative functions among different second messengers. This fine-tuning enables the organism to recruit an appropriate signaling molecule, and to decide between growth and sporulation.”.

[10] Liao CH, Xu Y, Rigali S, Ye BC. 2015. DasR is a pleiotropic regulator required for antibiotic production, pigment biosynthesis, and morphological development in *Saccharopolyspora erythraea*. *Appl Microbiol Biotechnol* 99:10215-10224.

- (8) Line 111. I suppose breakdown means metabolism? The whole section needs to be modified, as DasR represses GlcNAc metabolism and does not mediate its ‘breakdown’, more the

opposite. It prevents uptake and utilization of aminosugars.

Author response:

We apologize for the inaccurate terms. We have corrected it and modified the mentioned section as “The GntR-family regulator DasR functions as the developmental master regulator, is predominantly a dimer and a pleiotropic regulator that oversees the interplay between primary and secondary metabolism (20, 27, 28). In this study, we identified DasR of erythromycin-producing *Saccharopolyspora erythraea* as a target of c-di-AMP, thus revealing a direct link between c-di-AMP and GlcNAc signaling in this bacterium. We demonstrate that c-di-AMP binding stimulates DasR-mediated repression of GlcNAc uptake and utilization, and describe the molecular mechanism by which c-di-AMP mediates the formation of a c-di-AMP-linked DasR dimer. Such effective dimerization of DasR activates DasR-mediated regulation of its target regulon. Specifically, DAC activity is under direct transcriptional regulation by DasR, thus forming a regulatory circuit that governs the complete signaling cascade from nutritional status to the onset of morphological differentiation and antibiotic production. Finally, our phylogenetic analysis and *in vitro* assays further indicate that allosteric regulation by c-di-AMP exerts global regulation and signal integration mediated through DasR is likely conserved and essential among c-di-AMP-producing Actinobacteria.”.

Reviewer: 3 (Remarks to the Author):

This manuscript by You et. al. reports a novel receptor of the second messenger, cyclic di-AMP (cdA) in Actinobacteria. The authors generate a cdA overexpression strain, and observe differences in the utilization of N-acetylglucosamine (GlcNac), sporulation, and antibiotic production. The authors conclude that these differences are mediated by the central transcription factor, DasR, which was previously shown to bind and be inhibited by GlcNac. cdA, on the other hand, enhances the repression activity of DasR. Finally, DasR represses the cdA synthase, *disA*, suggesting that cdA negatively feedbacks on its own production. In sum, this is an interesting finding that expands what we know about cdA signaling. My biggest concern is the Kd of DasR binding to cdA was very high and likely not a physiologically relevant concentration of cdA and the concentrations of cdA used in the in vitro experiments, like the EMSAs, were quite high. Without showing the effects of cdA at more physiological concentrations, I have some doubts as to the validity of the model. This and other specific suggestions are listed below:

Author response:

We greatly appreciate the reviewer's positive comments. These comments are all valuable and very helpful for revising and improving our paper, as well as the important guiding significance to our work.

According to the reviewer's helpful suggestions, we have demonstrated this effect with lower concentrations of c-di-AMP. As shown in Figure S3A, the DNA-binding activity of DasR was enhanced with progressively increasing amounts of c-di-AMP, and further analysis using a relevant concentration of c-di-AMP (30 μ M) confirmed that the enhancement of DNA binding occurred with lower concentrations of c-di-AMP (Figure S3A). These figures were added to the revised manuscript.

In addition, we have changed the relative levels of c-di-AMP to absolute quantification. The c-di-AMP concentrations of the samples were normalized to the dry cell weight (Figure 1A). The intracellular concentrations of c-di-AMP were calculated using a cell volume of 7.14 mL/g dry cell weight, which was based on that of *E.coli*. The estimated c-di-AMP concentration in *S. erythraea* WT in rich medium is \sim 40 μ M. Moreover, in strains in which the c-di-AMP level was artificially increased 2 \sim 5-fold (Δ *dasR* or *OdisA*), suggesting that c-di-AMP binding by DasR should be relevant *in vivo*. The reported intracellular concentrations of c-di-AMP are approximately 0.5–70 μ M [1-3], including 28 μ M in *Synechococcus elongatus* [1] and 70 μ M in *Staphylococcus aureus* [2]. These numbers confirmed that c-di-AMP binding could occur under physiological conditions.

Figure S3. EMSA of DasR binding to its target gene *nagA* promoter. (A) Purified His-DasR (15 μ M) and a 300-bp DNA fragment were incubated with a gradient concentration of c-di-AMP (1, 5, 10, 30, 100, and 120 μ M). (B) Purified His-DasR and a 300-bp DNA fragment were

incubated without c-di-AMP (left) or with 30 μ M c-di-AMP (right).

Figure 1. (A) *disA* transcription levels and intracellular c-di-AMP concentrations in WT and *OdisA* strains grown in liquid TSB medium during late exponential growth (48 h). Fold change represents the expression level or c-di-AMP level compared to the WT strain. The c-di-AMP concentrations of the samples were normalized to the dry cell weight.

- [1] Gomelsky M, Rubin BE, Huynh TN, Welkie DG, Diamond S, Simkovsky R, Pierce EC, Taton A, Lowe LC, Lee JJ, Rifkin SA, Woodward JJ, Golden SS. 2018. High-throughput interaction screens illuminate the role of c-di-AMP in cyanobacterial nighttime survival. *PLoS Genet* 14:e1007301.
- [2] Rebecca, M., Corrigan, Lisa, Bowman, Alexandra, R., Willis, Volkhard, Kaefer. 2015. Cross-talk between Two Nucleotide-signaling Pathways in *Staphylococcus aureus*. *J Biol Chem* 290:5826-5839.
- [3] He J, Yin W, Galperin MY, Chou SH. 2020. Cyclic di-AMP, a second messenger of primary importance: tertiary structures and binding mechanisms. *Nucleic Acids Res* 48:2807-2829.

(1) Line 57, change “then” to “which”.

Author response:

Thank you for the comment. We have corrected it.

(2) Line 64-66-This is not a complete sentence. The manuscript is well written, but it could use some English editing to tighten up a few spots like this.

Author response:

We apologize for the mentioned inconvenience of the English language. According to the reviewer’s helpful suggestion, we have corrected the grammatical mistakes and carefully revised the whole manuscript with the help of a native English speaker.

We have corrected Line 64-66 as “Given the importance it plays in diverse bacterial physiology, the molecular mechanisms pertaining to nutrient metabolism and the related signal transduction are of increasing focus”.

(3) It would also be nice in the discussion if the authors could integrate their identification of DasR as a new receptor of c-di-AMP with the others that have been discovered. Specifically, is this an entirely new class or a new example of an existing class?

Author response:

Thank you for the helpful comment. DasR is a new example of an existing class. According to the reviewer's helpful suggestions, we have added this to the **Discussion** section as "C-di-AMP receptors are typically flexible in conformation, which enables a variety of physiological functions such as osmotic regulation, metabolic response and developmental regulation. Among the protein receptors of c-di-AMP, most are not of a single domain family, and the conservation of receptors appears to be relatively poor even among related organisms of the same phylum. In the current study, we showed that the activity of the master regulator DasR, also the GlcNAc-sensing regulator, is controlled by c-di-AMP. DasR is the first global regulator and a new example of GntR family regulator identified as the c-di-AMP receptor protein, which is quite distinct from the two c-di-AMP-binding transcriptional regulators (34, 35) characterized to date. DasR is a GntR member belongs to HutC subfamily whose members have been implicated in processes like antibiotic production and as sensors, of the nutritional status in the environment. Moreover, DasR homologs are widespread among hundreds of microorganisms synthesizing c-di-AMP. It is therefore likely that DasR represents a broadly conserved class of c-di-AMP receptors, bringing the regulatory role of this key second messenger into a new physiological arena in gram-positive multicellular Actinobacteria."

- (4) Fig. 1C-the authors need to specify which line is associated with which axis (solid for growth and dashed for utilization) in the figure legend.

Author response:

Thank you for the very helpful comments, we have added the mentioned description "Solid lines indicate growth and dashed lines illustrate GlcNAc utilization" to Fig. 1C in the revised manuscript.

- (5) Line 122-the authors should give some details here about how *disA* was overexpressed. What was the promoter used? Was an inducer added? Was this on a plasmid? Etc. And if a plasmid was used, was a vector control with inducer also present in the WT strain to control for non-specific effects of these components?

Author response:

Thank you for the questions. The vector pIB139 containing the *ermE* promoter was used for the expression of *disA*, and the *ermE* promoter is constitutive in *Saccharopolyspora erythraea* [16]. This vector has been verified to have no effects on the bacterial physiology [17]. These missing descriptions were added to the **Methods** section.

[16] Wilkinson CJ, Hughes-Thomas ZA, Martin CJ, Bohm I, Mironenko T, Deacon M, Wheatcroft M, Wirtz G, Staunton J, Leadlay PF. 2002. Increasing the efficiency of heterologous promoters in actinomycetes. *Journal of Molecular Microbiology and Biotechnology* 4:417-426.

[17] Xu Z, You D, Tang LY, Zhou Y, Ye BC. 2019. Metabolic Engineering Strategies Based on Secondary Messengers (p)ppGpp and C-di-GMP To Increase Erythromycin Yield in *Saccharopolyspora erythraea*. *ACS Synth Biol* 8:332-345.

- (6) Figure 1 legend-The authors should specify in this legend and other what statistical analysis was used to determine statistical significance.

Author response:

Thank you for the helpful comments. We have added the missing statements “Asterisks indicate a T-test significance value” for all the statistical analyses in the revised manuscript.

- (7) Fig. 2C-A Kd of 26.8 uM seems quite high as most cyclic di-nucleotides are present in low uM to nM concentrations in the cell. What is the physiological concentrations of cdA in Streptomyces?

Author response:

Thank you for the comments. Previous studies have reported that intracellular concentrations of c-di-AMP are approximately 0.5–70 μM [1-3], including 28 μM in *Synechococcus elongatus* [1] and 70 μM in *Staphylococcus aureus* [2]. The estimated c-di-AMP concentration in *S. erythraea* cells in rich medium is $\sim 40 \mu\text{M}$. Moreover, in strains in which the c-di-AMP level was artificially increased 2~5-fold (ΔdasR or OdisA), suggesting that c-di-AMP binding by DasR could occur under physiological conditions.

[1] Gomelsky M, Rubin BE, Huynh TN, Welkie DG, Diamond S, Simkovsky R, Pierce EC, Taton A, Lowe LC, Lee JJ, Rifkin SA, Woodward JJ, Golden SS. 2018. High-throughput interaction screens illuminate the role of c-di-AMP in cyanobacterial nighttime survival. PLoS Genet 14:e1007301.

[2] Rebecca, M., Corrigan, Lisa, Bowman, Alexandra, R., Willis, Volkhard, Kaefer. 2015. Cross-talk between Two Nucleotide-signaling Pathways in Staphylococcus aureus. J Biol Chem 290:5826-5839.

[3] He J, Yin W, Galperin MY, Chou SH. 2020. Cyclic di-AMP, a second messenger of primary importance: tertiary structures and binding mechanisms. Nucleic Acids Res 48:2807-2829.

- (8) Fig. 2D-cdA impacts shifting of DasR, but 100 uM is used, which is a very high amount that is not physiologically relevant. In order to support their conclusion, the authors should demonstrate this effect at lower concentration of cdA. Another way to strengthen these data is to demonstrate this effect is specific for cdA and such enhancement of DNA binding does not occur with other cdNs like c-di-GMP or cAMP.

Author response:

Thank you for the helpful suggestions. We have demonstrated this effect with lower concentrations of c-di-AMP. As shown in Figure S3A, the DNA-binding activity of DasR was enhanced with progressively increasing amounts of c-di-AMP, and further analysis using a relevant concentration of c-di-AMP (30 μM) confirmed that the enhancement of DNA binding occurred with lower concentrations of c-di-AMP (Figure S3B). These figures were added to the revised manuscript.

Figure S3. EMSA of DasR binding to its target gene *nagA* promoter. (A) Purified His-DasR (15 μM) and a 300-bp DNA fragment were incubated with a gradient concentration of c-di-AMP

(1, 5, 10, 30, 100, and 120 μM). (B) Purified His-DasR and a 300-bp DNA fragment were incubated without c-di-AMP (left) or with 30 μM c-di-AMP (right).

(9) Fig. 2G-What concentrations of cdA are used here?

Author response:

Thank you for the question. A gradient concentration of c-di-AMP (0, 30, and 100 μM) was used for Fig. 2G, and we have added this detail in the figure and figure legend.

Figure 2. (G) Crosslinking with a gradient concentration of c-di-AMP (0, 30, and 100 μM). Samples were analyzed by SDS-PAGE. Monomers and dimers are marked by arrows.

(10) Fig. 2G-the protein ladder should be indicated to show the size of the monomers and dimers.

Author response:

Thank you for the comments. The missing protein ladder of Fig. 2G was added in the revised manuscript.

Figure 2. (G) Crosslinking with a gradient concentration of c-di-AMP (0, 30, and 100 μM). Samples were analyzed by SDS-PAGE. Monomers and dimers are marked by arrows.

(11) Fig. 3E-A nice control for this experiment would have been to demonstrate that the cdA repression of these genes does not occur in a *dasR* mutant. In other words, analysis of a *dasR* versus a *dasR/OdisA* should not demonstrate cdA repression.

Author response:

Thank you for the very helpful comments. We have constructed a $\Delta*dasR*::*disA*$ strain and added the mentioned analysis as shown in revised Figure 3E. The results showed that c-di-AMP had no effect on the transcription of GlcNAc assimilation-related genes in the *dasR*-deficient strain ($\Delta*dasR*::*disA*$), supporting the idea that c-di-AMP through DasR controls the GlcNAc assimilation.

Figure 3. (E) The transcription levels of the indicated genes in *S. erythraea* WT, *OdisA*, Δ *dasR* and Δ *dasR::disA* strains grown in liquid TSB medium during late exponential growth (48 h). Fold change represents the expression level compared to the WT strain. Error bars show the SDs of three independent experiments. Asterisks indicate a T-test significance value. * $P < 0.05$, ** $P < 0.01$, *** $P < 0.001$, **** $P < 0.0001$.

(12) Fig. 3b-I am confused by what this represents. Is this the summary of all ChIP-seq signals or a representative peak?

Author response:

Thank you for the questions. Yes, Fig. 3B represents the summary of all ChIP-seq signals.

(13) Fig. 4C-The authors use a *dasR* deletion strain, but in lines 118 and 302 they state that this gene is essential, which means that you cannot generate a null mutation and still have viability. Could the authors clarify.

Author response:

We apologize for the inappropriate choice of word. We have changed “essential” to “crucial”.

(14) Lines 353-356-What the authors are describing here is really a negative feedback loop on *cdA* synthesis to limit overproduction of this signal.

Author response:

Thank you for the comments. Yes, it is a negative feedback loop and we have added this conclusion to the revised manuscript.

(15) Fig. 6-One central question that I was left with and was unaddressed is how *cdA* through *DasR* leads to enhanced sporulation and antibiotic production. Is *DasR* not functioning as a repressor for these genes or is there an intermediate regulator. Does *DasR* directly bind to the promoters of these genes in the ChIP-seq experiment. Some discussion around lines 371-372 is warranted.

Author response:

We apologize for the confusion. *DasR* is indeed a pleiotropic regulator since it is a repressor of GlcNAc metabolism and an activator of antibiotic production and morphological development in *Saccharopolyspora erythraea* [9,10]. Hence, c-di-AMP binding causes hyper-repression of GlcNAc metabolism and activates *DasR*-mediated developmental transition and antibiotic production. We have revised this according to the reviewer’s helpful suggestion. Yes, ChIP-seq enables genome-wide mapping of transcription factor binding and the revelation of underlying molecular mechanisms of differential gene regulation that are

governed by specific transcription factors. DasR directly bound to the promoters of these genes in the ChIP-seq experiment. For the mentioned warrant of lines 371-372, we have added the missing references to the revised manuscript.

- [9] Liao CH, Rigali S, Cassani CL, Marcellin E, Nielsen LK, Ye BC. 2014. Control of chitin and N-acetylglucosamine utilization in *Saccharopolyspora erythraea*. *Microbiology-Sgm* 160:1914-1928.
- [10] Liao CH, Xu Y, Rigali S, Ye BC. 2015. DasR is a pleiotropic regulator required for antibiotic production, pigment biosynthesis, and morphological development in *Saccharopolyspora erythraea*. *Appl Microbiol Biotechnol* 99:10215-10224.

REVIEWER COMMENTS

Reviewer #1 (Remarks to the Author):

The revised paper looks very nice. The authors have responded to my comments and provided quantitative data for the proposed interaction between DasR and c-di-GMP. Unfortunately, to justify the relatively low affinity of DasR towards c-di-AMP (K_d value as high as 26.8 μM), the authors included in their response a wrong and misleading statement "Previous studies have reported that intracellular concentrations of c-di-AMP are approximately 0.5–70 μM [1-3], including 28 μM in *Synechococcus elongatus* [1] and 70 μM in *Staphylococcus aureus* [2]".

I have carefully checked all three references and neither of them lists the 28 μM value, let alone the 70 μM value. The highest value that is shown in ref. [1] for *Synechococcus elongatus* is 18.8 μM (which is also cited in Ref. [3]). Ref. [2] measured the c-di-AMP levels in *Staphylococcus aureus* and concluded that "Bacteria in the exponential growth phase (2-h time point) contained 2.43 ± 1.87 μM c-di-AMP, and the concentration increased to 8.09 ± 0.96 μM at the 4-h time point when cells reached the late exponential/early stationary growth phase and remained high in the stationary phase (Fig. 1C)." Ref. 2 does show a higher value of (54.93 ± 8.15 μM c-di-AMP) in the LAC*ΔgdpP mutant strain that lacks a c-di-AMP phosphodiesterase but even in that mutant, c-di-AMP levels never reach the 70 μM value.

It is important to note that these figures, 28 μM for *Synechococcus elongatus* and 70 μM for *Staphylococcus aureus*, do not appear anywhere in the manuscript. As a result, I believe that, in their response, the authors deliberately misled the reviewers and the journal editor. Accordingly, their otherwise nicely written paper must be rejected and the authors should be instructed to never do that again.

Reviewer #2 (Remarks to the Author):

The authors have added a lot of new data, which has significantly improved the manuscript. This makes the data in the paper a lot more solid than in the previous version. The new ITC (Fig. S2) as well as EMSA data (Fig. S3) are very useful and shed more light on the effect and specificity of cda on DasR binding. Other new experiments that improved the paper a lot are the absolute quantification of cda levels in the cell and the details on growth conditions.

An issue that remains is the fact that the data only relate to *Sacch erythraea*. Ref #1 states "The paper would greatly benefit from a comparison of those DasRs that are expected to bind c-di-AMP and those that are not". This is a crucial point. After all, many Actinobacteria (some 40%) do not produce cda, while

they do have DasR. The authors do not really address this major issue and simply edit the text a bit. This is not sufficient. Secondly, there is the surprising difference between *S. coelicolor* and *Sacch. erythraea* (see Discussions with Ref#2), considering the generally high conservation of DasR. I believe the data are solid enough to at least prove the validity of the interactions for the chosen system (*Sacch. erythraea*), but that needs to be made clear. Hence, add "*Saccharopolyspora erythraea*" to the title and also mention in the abstract that the work only applies to streptomycetes that produce *cda*.

All in all, I believe it is very important to perform the same EMSAs/ITC studies using purified DasR from several other Actinobacteria and in particular those that do not produce *cda*, and in addition DasR from *S. coelicolor*, to study the binding of *cda* to DasR and its effect on DNA binding. This would add valuable data on the host specificity of the effect of *cda*. And may explain at least in part the major difference seen between the role of DasR in growth and antibiotic production in *S. coelicolor* and *Sacch. erythraea*.

As a minor comment, replace the occurrences of "actinomycetes" by "Actinobacteria", namely in line 315 and in Figure 5.

Reviewer #3 (Remarks to the Author):

This revised manuscript by You et. al. has addressed all of my concerns.

Reviewer #4 (Remarks to the Author):

The manuscript by Di You and colleagues explores the interaction between the actinobacterial GntR-family transcriptional regulator DasR and the second messenger *c*-di-AMP. In agreement with the previous reviewers' comments, the authors present convincing genetic data supporting the hypothesis that *c*-di-AMP binding to DasR is crucial in the N-acetylglucosamine (GlcNAc) signaling pathway. This research contributes significantly to our understanding of *c*-di-AMP homeostasis and its regulatory functions in Actinobacteria. The manuscript gains additional credibility by including additional experiments and supplementary data in response to previous reviewers' comments. However, I would recommend publication after the following questions are addressed.

1) In response to Reviewer #1's second question, the authors applied molecular docking rather than experimental methods to characterize the complex structure of *c*-di-AMP-DasR. The statement that 'binding of *c*-di-AMP occurred in the effector-binding domain of DasR' is based solely on the presented

docking results with no experimental validation (e.g., mutagenesis experiments). Therefore, this statement is not convincing and should be revised.

2) The 'Molecular Docking Analysis' section of the Methods lacks detailed parameters, such as the size of the grid box used in AutoDock. These parameters are crucial for evaluating the reliability of the docking study and for reproducing the results by others.

3) Lines 547-548: "The best poses of ligand-receptor complexes ..." Is the pose displayed in Fig. S4 the top pose? Please provide the ranking No. and the AutoDock score for this pose in the manuscript.

4) Fig. S4: Please display the modeled DasR dimer in different colors for a clearer view of the interface.

5) Can you determine the stoichiometry of the c-di-AMP : DasR dimer complex?

6) Lines 112 – 116: "Such effective dimerization of DasR activates DasR-mediated regulation of its target regulon. Specifically, DAC activity is under direct transcriptional regulation by DasR, thus forming a regulatory circuit that governs the complete signaling cascade from nutritional status to the onset of morphological differentiation and antibiotic production." Explain why dimerization helps in the manuscript.

Minor Concerns:

1) Line 544: 'Auto Dock Pdbqt' should be 'AutoDock PDBQT'.

2) Line 545: 'AutoDock (version 1.4.6)'. Do you mean the version of MGLTools?

3) Lines 547-549: 'The best poses of ligand-receptor complexes of hydrogen bonds and electrostatic interactions were expressed as binding energy values (kcal/mol) to represent docking results.' This sentence is confusing and should be polished.

4) Line 550: '... atoms binding to the ligands and receptors ...'. Do you mean "atoms involved in ligand-receptor binding"?

Response to the reviewers

Reviewers' comments:

Reviewer: 1 (Remarks to the Author):

The revised paper looks very nice. The authors have responded to my comments and provided quantitative data for the proposed interaction between DasR and c-di-AMP. Unfortunately, to justify the relatively low affinity of DasR towards c-di-AMP (K_d value as high as 26.8 μM), the authors included in their response a wrong and misleading statement "Previous studies have reported that intracellular concentrations of c-di-AMP are approximately 0.5–70 μM [1-3], including 28 μM in *Synechococcus elongatus* [1] and 70 μM in *Staphylococcus aureus* [2]".

I have carefully checked all three references and neither of them lists the 28 μM value, let alone the 70 μM value. The highest value that is shown in ref. [1] for *Synechococcus elongatus* is 18.8 μM (which is also cited in Ref. [3]). Ref. [2] measured the c-di-AMP levels in *Staphylococcus aureus* and concluded that "Bacteria in the exponential growth phase (2-h time point) contained 2.43 ± 1.87 μM c-di-AMP, and the concentration increased to 8.09 ± 0.96 μM at the 4-h time point when cells reached the late exponential/early stationary growth phase and remained high in the stationary phase (Fig. 1C)." Ref. 2 does show a higher value of (54.93 ± 8.15 μM c-di-AMP) in the LAC*ΔgdpP mutant strain that lacks a c-di-AMP phosphodiesterase but even in that mutant, c-di-AMP levels never reach the 70 μM value.

It is important to note that these figures, 28 μM for *Synechococcus elongatus* and 70 μM for *Staphylococcus aureus*, do not appear anywhere in the manuscript. As a result, I believe that, in their response, the authors deliberately misled the reviewers and the journal editor. Accordingly, their otherwise nicely written paper must be rejected and the authors should be instructed to never do that again.

Author response:

We greatly appreciate your positive comments and the thoughtful suggestions for revising and improving our paper.

We sincerely apologize for the lack of a detailed explanation of the calculation processes of the mentioned intracellular c-di-AMP in the previous response. We have carefully revised the text and added the necessary illustrations to prevent confusion, which we hope will be met with approval. In our former sentence, "Previous studies have also reported that intracellular concentrations of c-di-AMP are approximately 0.5–70 μM [1-3], including 28 μM in *Synechococcus elongatus* [1] and 70 μM in *Staphylococcus aureus* [2]", we intended to summarize the maxima of intracellular c-di-AMP reported in previous research. The maximum concentrations of intracellular c-di-AMP in *Synechococcus elongatus* and *Staphylococcus aureus* were calculated by means of point conversion. The maximum (28 μM) concentration of c-di-AMP in *Synechococcus elongatus* was consistent with the WT value of 18.8 μM (Fig. R1B) after a ~1.5-fold increase, which corresponds to the value of approximately 28 μM observed at night (Fig. R1C, the 24 h time point)^[1]. For *Staphylococcus aureus*, the maximum

concentration of 70 μM is attributed to the observed increase in c-di-AMP levels in the *Staphylococcus aureus* LAC* ΔgdpP strain at the 2 h time point during bacterial growth (Fig. R2B) [2].

We admire your meticulous scholarship and learned a great deal in the process of revision.

Figure R1. Original Fig 1 and the figure legend in reference [1].

[REDACTED]

Figure R2. Original FIGURE 3 and the figure legend in reference [2].

[REDACTED]

[1] Gomelsky M, Rubin BE, Huynh TN, Welkie DG, Diamond S, Simkovsky R, Pierce EC, Taton A, Lowe LC, Lee JJ, Rifkin SA, Woodward JJ, Golden SS. 2018. High-throughput interaction screens illuminate the role of c-di-AMP in cyanobacterial nighttime survival. *PLoS Genet* 14:e1007301.

[2] Rebecca, M., Corrigan, Lisa, Bowman, Alexandra, R., Willis, Volkhard, Kaefer. 2015. Cross-talk between Two Nucleotide-signaling Pathways in *Staphylococcus aureus*. *J Biol Chem* 290:5826-5839.

Reviewer: 2 (Remarks to the Author):

The authors have added a lot of new data, which has significantly improved the manuscript. This makes the data in the paper a lot more solid than in the previous version. The new ITC (Fig. S2) as well as EMSA data (Fig. S3) are very useful and shed more light on the effect and specificity of *cda* on DasR binding. Other new experiments that improved the paper a lot are the absolute quantification of *cda* levels in the cell and the details on growth conditions.

An issue that remains is the fact that the data only relate to *Sacch. erythraea*. Ref #1 states "The paper would greatly benefit from a comparison of those DasRs that are expected to bind c-di-AMP and those that are not". This is a crucial point. After all, many Actinobacteria (some 40%) do not produce *cda*, while they do have DasR. The authors do not really address this major issue and simply edit the text a bit. This is not sufficient. Secondly, there is the surprising difference between *S. coelicolor* and *Sacch. erythraea* (see Discussions with Ref#2), considering the generally high conservation of DasR. I believe the data are solid enough to at least prove the validity of the interactions for the chosen system (*Sacch. erythraea*), but that needs to be made clear. Hence, add "*Saccharopolyspora erythraea*" to the title and also mention in the abstract that the work only applies to Streptomycetes that produce *cda*.

All in all, I believe it is very important to perform the same EMSAs/ITC studies using purified DasR from several other Actinobacteria and in particular those that do not produce *cda*, and in addition DasR from *S. coelicolor*, to study the binding of *cda* to DasR and its effect on DNA binding. This would add valuable data on the host specificity of the effect of *cda*. And may explain at least in part the major difference seen between the role of DasR in growth and antibiotic production in *S. coelicolor* and *Sacch. erythraea*.

Author response:

We greatly appreciate your positive comments, as well as the time and the effort spent reviewing our work. These comments were very valuable for revising and improving our paper. According to your helpful comments, we performed the same EMSA/ITC studies using purified DasR from *S. coelicolor*. As shown in Fig. R3, a specific interaction between c-di-AMP and the DasR^{Sc} protein was also observed, with a KD of 23.0 μM (Fig. R3A). We also performed ITC assays using DasR^{Sc} titrated with cAMP, cGMP, and c-di-GMP, and the results showed that none of the detected nucleotides bind DasR^{Sc} (Fig. R3B-D). These observations support a specific interaction with c-di-AMP. In addition, the EMSA (Fig. 5G) led to the same conclusion as the one using DasR from *Sacch. erythraea*: c-di-AMP strongly induced DasR^{Sc} binding to DNA. Taken together, these results show that DasR is a bona fide bacterial c-di-AMP receptor protein and support a model whereby c-di-AMP enhances the DNA-binding activity of DasR in *S. coelicolor*. These figures were added to the revised manuscript. The title and the abstract were also revised according to your suggestions.

Figure R3. Characterization of the interactions of nucleotides with DasR^{Sco} using ITC. Titration of nucleotides (1 mM) into DasR^{Sco} (10 μM). ITC measurements of DasR^{Sco} titrated with c-di-AMP (A), cAMP (B), cGMP (C), and c-di-GMP (D).

Figure 5. (G) EMSA of DasR^{Sco} binding to its target gene *disA* promoter. Purified His-DasR^{Sco} and DNA fragment were incubated with 30 μM c-di-AMP (bottom) or without c-di-AMP (top).

Regarding the abovementioned measurements in several other Actinobacteria, in particular those that do not produce *cda*, we apologize that we could not complete them in the present study due to the lack of information on DasR, *dre* and the targets in these bacteria. We fully agree that these findings would be valuable for determining the host specificity of the effect of c-di-AMP. Your suggestion provides a direction for our next research. We have discussed this point in the revised manuscript, and we will focus on exploring these questions in our future research.

As a minor comment, replace the occurrences of "actinomycetes" by "Actinobacteria", namely in line 315 and in Figure 5.

Author response:

Thank you for your comments. We have corrected it.

Reviewer: 3 (Remarks to the Author):

This revised manuscript by You et. al. has addressed all of my concerns.

Author response:

We greatly appreciate for your positive remarks.

Reviewer: 4 (Remarks to the Author):

The manuscript by Di You and colleagues explores the interaction between the actinobacterial GntR-family transcriptional regulator DasR and the second messenger c-di-AMP. In agreement with the previous reviewers' comments, the authors present convincing genetic data supporting the hypothesis that c-di-AMP binding to DasR is crucial in the N-acetylglucosamine (GlcNAc) signaling pathway. This research contributes significantly to our understanding of c-di-AMP homeostasis and its regulatory functions in Actinobacteria. The manuscript gains additional credibility by including additional experiments and supplementary data in response to previous reviewers' comments. However, I would recommend publication after the following questions are addressed.

Author response:

We greatly appreciate your positive comments. These comments are very valuable for revising and improving our paper, as well as the important guiding significance to our work.

- (1) In response to Reviewer #1's second question, the authors applied molecular docking rather than experimental methods to characterize the complex structure of c-di-AMP-DasR. The statement that 'binding of c-di-AMP occurred in the effector-binding domain of DasR' is based solely on the presented docking results with no experimental validation (e.g., mutagenesis experiments). Therefore, this statement is not convincing and should be revised.

Author response:

Thank you for your helpful comment. We performed the necessary mutagenesis experiments. Generally, the helix-turn-helix (HTH) motif mediates DNA binding. To probe the binding mechanism further, we created mutations that retained binding at the HTH motif but disabled binding at the effector-binding (EB) motif (Fig. S4C). Based on our data (Fig. S4D), DasR-HTH bound to DNA, whereas no reliable DNA-binding activity was detected for DasR-EB. We also observed that an EB deletion mutant of DasR was incapable of superbinding in the presence of c-di-AMP (Fig. S4D). Thus, the effector-binding motif of DasR appears to be the main motif responsible for its ability to bind c-di-AMP. The relevant figures were added to the revised manuscript.

Figure S4. The putative c-di-AMP-DasR complex. (A) Model of DasR dimer obtained from AlphaFold2 compared with 4ZS8. (B) the putative DasR-c-di-AMP docking (ranking No.1 with a binding energy of -6.81 kcal/mol). The close-up shows the predicted interaction sites, annotated with the putative binding residues. Interactions are denoted with labels. (C) Construction of the DasR HTH (DasR¹⁻¹⁰⁰) and DasR EB (DasR¹⁰⁰⁻²⁵¹) mutants. (D) EMSA of DasR mutants binding to its target gene *nagA* promoter. Purified His-DasR mutants and DNA fragment were incubated with 100 μM c-di-AMP (bottom) or without c-di-AMP (top).

- (2) The ‘Molecular Docking Analysis’ section of the Methods lacks detailed parameters, such as the size of the grid box used in AutoDock. These parameters are crucial for evaluating the reliability of the docking study and for reproducing the results by others.

Author response:

We apologize for the missing details. According to your helpful suggestion, we have added the missing information to the revised manuscript as follows: “.... The docking center coordinates X, Y, and Z were set as follows: center_x = -5.907, center_y = -0.118, and center_z = 6.553. All dimensions of the docking box were divided into a grid of 122*84*110 points with a grid spacing of 0.375 Å. The maximum limit when searching for conformations was set to 2000, and the genetic algorithm was used to sample and score conformations via semiflexible

docking. The optimal configurations of ligand-receptor hydrogen bond complexes and electrostatic interactions were determined in terms of binding energy (kcal/mol) to reflect the outcomes of docking.... ”.

- (3) Lines 547-548: “The best poses of ligand-receptor complexes ...” Is the pose displayed in Fig. S4 the top pose? Please provide the ranking No. and the AutoDock score for this pose in the manuscript.

Author response:

Thank you for your question. Yes, the displayed pose of the DasR-c-di-AMP complex was ranked No. 1, with a binding energy of -6.81 kcal/mol. We have added this missing information to the revised manuscript.

- (4) Fig. S4: Please display the modeled DasR dimer in different colors for a clearer view of the interface.

Author response:

Thank you for your helpful comments. We have changed the color scheme of Fig. S4 in the revised manuscript.

Figure S4. The putative c-di-AMP-DasR complex. (A) Model of DasR dimer obtained from AlphaFold2 compared with 4ZS8. (B) the putative DasR-c-di-AMP docking (ranking No.1 with a binding energy of -6.81 kcal/mol). The close-up shows the predicted interaction sites, annotated with the putative binding residues. Interactions are denoted with labels.

(5) Can you determine the stoichiometry of the c-di-AMP: DasR dimer complex?

Author response:

Thank you for your question. We fully agree that determining the stoichiometry of the c-di-AMP-DasR complex would be useful for understanding the details of its interaction and enhancement. In fact, we have been trying to resolve the crystal structure of the c-di-AMP-DasR complex, but unfortunately, we have not yet obtained the suitable crystals. We are terribly sorry that we could not determine the stoichiometry of the c-di-AMP-DasR complex in the present study. For purposes of molecular docking, we assumed that the stoichiometric ratio was 1:1 (the most common binding stoichiometry). We have added this limitation to the Discussion section, and future studies looking further into the structural determination of the actual binding mode of c-di-AMP in DasR will be critical next steps in this work.

(6) Lines 112 – 116: “Such effective dimerization of DasR activates DasR-mediated regulation of its target regulon. Specifically, DAC activity is under direct transcriptional regulation by DasR, thus forming a regulatory circuit that governs the complete signaling cascade from nutritional status to the onset of morphological differentiation and antibiotic production.” Explain why dimerization helps in the manuscript.

Author response:

Thank you for your helpful comments. We have added the missing explanation “... Since GntR-family regulators interact with DNA as dimers (31), such effective dimerization of DasR activates DasR-mediated regulation of its target regulon. Specifically, DAC activity is under direct transcriptional regulation by DasR, thus forming a regulatory circuit that governs the complete signaling cascade from nutritional status to the onset of morphological differentiation and antibiotic production.” in the revised manuscript.

(7) Line 544: ‘Auto Dock Pdbqt’ should be ‘AutoDock PDBQT’.

Author response:

Thank you for your comment. We have corrected it.

(8) Line 545: ‘AutoDock (version 1.4.6)’. Do you mean the version of MGLTools?

Author response:

We apologize for the mistake. Yes, we have corrected it as “MGLTools (version 1.4.6)”.

(9) Lines 547-549: ‘The best poses of ligand-receptor complexes of hydrogen bonds and electrostatic interactions were expressed as binding energy values (kcal/mol) to represent docking results.’ This sentence is confusing and should be polished.

Author response:

Thank you for your comment. We have revised it as “The optimal configurations of ligand-receptor hydrogen bond complexes and electrostatic interactions were determined in terms of binding energy (kcal/mol) to reflect the outcomes of docking.”.

(10) Line 550: ‘... atoms binding to the ligands and receptors ...’. Do you mean “atoms involved in ligand-receptor binding”?

Author response:

Thank you for your question. Yes, we have corrected it as “atoms involved in ligand-receptor binding”.

REVIEWERS' COMMENTS

Reviewer #1 (Remarks to the Author):

The manuscript by You and colleagues went through two cycles of revision, which substantially improved the text and presentation of the results. The authors used a variety of approaches to support their idea that a) the transcriptional regulator DasR interacts with the second messenger c-di-AMP and b) this interaction affects binding of DasR to its regulator N-acetylglucosamine (GlcNac). The only remaining point of contention is the relatively low affinity of DasR for c-di-AMP. In order for the proposed interaction of DarS with c-di-AMP to occur at the normal cellular levels of both molecules and, accordingly, to represent a physiologically relevant mechanism, the studied organisms, *Saccharopolyspora erythraea* and *Streptomyces coelicolor*, must have much higher intracellular c-di-AMP levels than any organisms studied before.

Indeed, as mentioned previously, the reported affinity of DasR for c-di-AMP ($K_D = 26.8 \mu\text{M}$ in *S. erythraea*, Fig. 2C, and $K_D = 23.0 \mu\text{M}$ in *S. coelicolor*, Fig. 5F) is far lower than the values reported for known cellular c-di-AMP receptor, which are typically in submicromolar range (see He et al. 2020, ref. 6 in the reviewed paper). Also, Fig. 2B shows no response (misspelled!) upon the addition of $10 \mu\text{M}$ c-di-AMP. Several c-di-AMP receptors have been reported to have K_D s as high as high as $10 \mu\text{M}$ but $26.8 \mu\text{M}$ is still much higher than any K_D value reported so far. Unfortunately, in Fig. 1A and Fig. 4D, cellular c-di-AMP levels in wild-type cells are expressed as $\sim 300 \text{ nmol/mg}$ dry weight, which cannot be easily converted into an intracellular concentration. It would have been helpful to calculate intracellular cell volumes and convert nmol/mg dry weight into actual cellular concentrations.

Minor comments

L. 41. After "Interestingly" add "the expression of"

L. 45. 'Streptomycetes' should not be capitalized.

L. 53-54. Remove "that do not use c-di-GMP". This statement, taken from ref. 1 of the reviewed paper, is wrong and misleading: there are very few gram-positive bacteria that use c-di-AMP but not c-di-GMP, particularly among prominent human pathogens and [organisms] of environmental importance (see ref. 2 and 10).

Reviewer #2 (Remarks to the Author):

I have carefully read the manuscript again, and it is clear that the authors have addressed all the comments by both reviewers very well. It is very interesting to see the specificity of c-di-AMP for DasR. This means there is a possible myriad of compounds that affect its activity. One wonders which molecules accumulate when during development and how this balance affects DasR activity, but I agree that this is beyond the scope of this work. I am satisfied with the response to my previous comments and thank the authors for their efforts.

Reviewer #3 (Remarks to the Author):

The authors have addressed my concerns.

Reviewer #4 (Remarks to the Author):

The revised manuscript is improved. The authors have responded to my previous comments and presented additional data. However, I still have several questions that require clarification from the authors.

1. As molecular docking was employed to characterize the c-di-AMP-DasR complex structure, it is crucial to validate the binding site of c-di-AMP in DasR through site-directed mutagenesis. Specifically, mutations should be introduced into the key residues (e.g., at least two single mutations selected from residues 238-248) identified by the docking study to assess their impact on c-di-AMP binding, such as changes in the K_d value. This experimental validation would strengthen the conclusions obtained from the docking analysis.

2. The c-di-AMP binding site of DasR provided by the authors (Figure S4) seems to be on the dimer interface of DasR. Please elaborate on the mechanism by which c-di-AMP induces the dimerization of DasR based on this binding mode.

3. Lines 568-570: "The optimal configurations of ligand-receptor hydrogen bond complexes and electrostatic interactions were determined in terms of binding energy (kcal/mol) to reflect the outcomes

of docking.” Do you mean “The best pose of c-di-AMP was determined based on binding energy (kcal/mol) to reflect the outcomes of docking”?

4. Line 557: Both the AlphaFold2 model and the crystal structure of DasR were prepared. Please clarify which structure was utilized for docking and provide the rationale of the choice.

Response to the reviewers

Reviewers' comments:

Reviewer: 1 (Remarks to the Author):

The manuscript by You and colleagues went through two cycles of revision, which substantially improved the text and presentation of the results. The authors used a variety of approaches to support their idea that a) the transcriptional regulator DasR interacts with the second messenger c-di-AMP and b) this interaction affects binding of DasR to its regulator N-acetylglucosamine (GlcNAc). The only remaining point of contention is the relatively low affinity of DasR for c-di-AMP. In order for the proposed interaction of DarR with c-di-AMP to occur at the normal cellular levels of both molecules and, accordingly, to represent a physiologically relevant mechanism, the studied organisms, *Saccharopolyspora erythraea* and *Streptomyces coelicolor*, must have much higher intracellular c-di-AMP levels than any organisms studied before.

Indeed, as mentioned previously, the reported affinity of DasR for c-di-AMP ($K_D = 26.8 \mu\text{M}$ in *S. erythraea*, Fig. 2C, and $K_D = 23.0 \mu\text{M}$ in *S. coelicolor*, Fig. 5F) is far lower than the values reported for known cellular c-di-AMP receptor, which are typically in submicromolar range (see He et al. 2020, ref. 6 in the reviewed paper). Also, Fig. 2B shows no response (misspelled!) upon the addition of $10 \mu\text{M}$ c-di-AMP. Several c-di-AMP receptors have been reported to have K_D s as high as high as $10 \mu\text{M}$ but $26.8 \mu\text{M}$ is still much higher than any K_D value reported so far. Unfortunately, in Fig. 1A and Fig. 4D, cellular c-di-AMP levels in wild-type cells are expressed as $\sim 300 \text{ nmol/mg}$ dry weight, which cannot be easily converted into an intracellular concentration. It would have been helpful to calculate intracellular cell volumes and convert nmol/mg dry weight into actual cellular concentrations.

Author response:

We fully agree with Reviewer 1 and appreciate this critical comment for revising and improving our paper. The intracellular concentrations of c-di-AMP were calculated using a cell volume of 7.14 mL/g dry cell weight, which was based on that of *E. coli*. The estimated c-di-AMP concentration in the *S. erythraea* WT strain was $\sim 40 \mu\text{M}$. Moreover, in strains in which the c-di-AMP level was artificially increased 2~5-fold (ΔdasR or OdisA), these numbers confirmed that c-di-AMP binding could occur under physiological conditions. The relevant figures have been revised accordingly. The misspelled words in Fig. 2 have also been corrected in the revised manuscript.

Minor comments

(1) L. 41. After "Interestingly" add "the expression of"

Author response:

Thank you for your comment. We have added this phrase.

L. 45. 'Streptomyces' should not be capitalized.

Author response:

Thank you for your comment, we have corrected this.

L. 53-54. Remove "that do not use c-di-GMP". This statement, taken from ref. 1 of the reviewed paper, is wrong and misleading: there are very few gram-positive bacteria that use c-di-AMP but not c-di-GMP, particularly among prominent human pathogens and [organisms] of environmental importance (see ref. 2 and 10).

Author response:

Thank you for this important comment. We have deleted this phrase.

Reviewer: 4 (Remarks to the Author):

The revised manuscript is improved. The authors have responded to my previous comments and presented additional data. However, I still have several questions that require clarification from the authors.

Author response:

We greatly appreciate your constructive comments. All the comments and concerns are thoroughly addressed in our revised manuscript and were greatly welcomed, as they have significantly improved the quality of our findings.

1. As molecular docking was employed to characterize the c-di-AMP-DasR complex structure, it is crucial to validate the binding site of c-di-AMP in DasR through site-directed mutagenesis. Specifically, mutations should be introduced into the key residues (e.g., at least two single mutations selected from residues 238-248) identified by the docking study to assess their impact on c-di-AMP binding, such as changes in the K_d value. This experimental validation would strengthen the conclusions obtained from the docking analysis.

Author response:

We fully agree that validating the binding sites identified by the docking study would be useful for understanding the details of the c-di-AMP-DasR interaction. To perform a more systematic study of the c-di-AMP-DasR interaction, we are conducting further docking analysis and site-directed mutagenesis experiments. We will show the results in our next paper.

2. The c-di-AMP binding site of DasR provided by the authors (Figure S4) seems to be on the dimer interface of DasR. Please elaborate on the mechanism by which c-di-AMP induces the dimerization of DasR based on this binding mode.

Author response:

Thank you for this helpful suggestion. We have added this description as “The putative DasR-c-di-AMP docking analysis (Supplementary Fig. 4a, b) suggested that c-di-AMP acted as a dimerizer to link two DasR protomers to drive DasR dimerization.”

3. Lines 568-570: “The optimal configurations of ligand-receptor hydrogen bond complexes and electrostatic interactions were determined in terms of binding energy (kcal/mol) to reflect the outcomes of docking.” Do you mean “The best pose of c-di-AMP was determined based on binding energy (kcal/mol) to reflect the outcomes of docking”?

Author response:

Thank you for your question. Yes, we have corrected it to “The best pose of c-di-AMP was determined based on binding energy (kcal/mol) to reflect the outcomes of docking”.

4. Line 557: Both the AlphaFold2 model and the crystal structure of DasR were prepared. Please clarify which structure was utilized for docking and provide the rationale of the choice.

Author response:

Thank you for your comments. We have clarified this point as “The three-dimensional (3D) structure of *S. erythraea* DasR was predicted in AlphaFold2. The structure of *S. coelicolor* DasR

(PDB; 4ZS8 (46)) was retrieved from PDB. The structure of c-di-AMP was retrieved from PubChem. Molecular docking was performed on *S. erythraea* DasR with c-di-AMP...